# Genome-scale annotation of protein binding sites via language model and geometric deep learning

**Qianmu Yuan, Chong Tian, Yuedong Yang***

School of Computer Science and Engineering, Sun Yat-sen University, Guangzhou, China

**Abstract** Revealing protein binding sites with other molecules, such as nucleic acids, peptides, or small ligands, sheds light on disease mechanism elucidation and novel drug design. With the explosive growth of proteins in sequence databases, how to accurately and efficiently identify these binding sites from sequences becomes essential. However, current methods mostly rely on expensive multiple sequence alignments or experimental protein structures, limiting their genome-scale applications. Besides, these methods haven't fully explored the geometry of the protein structures. Here, we propose GPSite, a multi-task network for simultaneously predicting binding residues of DNA, RNA, peptide, protein, ATP, HEM, and metal ions on proteins. GPSite was trained on informative sequence embeddings and predicted structures from protein language models, while comprehensively extracting residual and relational geometric contexts in an end-to-end manner. Experiments demonstrate that GPSite substantially surpasses state-of-the-art sequence-based and structure-based approaches on various benchmark datasets, even when the structures are not well-predicted. The low computational cost of GPSite enables rapid genome-scale binding residue annotations for over 568,000 sequences, providing opportunities to unveil unexplored associations of binding sites with molecular functions, biological processes, and genetic variants. The GPSite webserver and annotation database can be freely accessed at https://bio-web1.nscc-gz.cn/app/GPSite.

**\*For correspondence:** yangyd25@mail.sysu.edu.cn

**Competing interest:** The authors declare that no competing interests exist.

**Sent for Review** 02 November 2023
**Preprint posted** 05 November 2023
**Reviewed preprint posted** 11 January 2024
**Reviewed preprint revised** 26 March 2024
**Version of Record published** 17 April 2024

## eLife assessment

The authors introduce a **valuable** machine-learning model for predicting binding sites of diverse ligands, including DNA, RNA, peptides, proteins, ATP, HEM, and metal ions, on proteins. The method is freely accessible and user-friendly. The authors have conducted thorough benchmarking and ablation studies, providing **convincing** evidence of the model's overall performance, despite some imperfections of the comparisons to other methods that arise from intrinsic differences between training methods and data.

## Introduction

Proteins perform most biological functions by specifically interacting with other molecules such as nucleic acids, peptides, proteins, or small ligands of various kinds (*Alipanahi et al., 2015*; *Rolland et al., 2014*; *Andreini et al., 2009*). Knowledge of these binding interfaces benefits protein function prediction, disease mechanism understanding, and novel drug design (*Lee et al., 2007*; *Wang et al., 2017*; *Wells and McClendon, 2007*). Although experimental techniques such as X-ray crystallography and nuclear magnetic resonance spectroscopy can solve the native complex structures to detect binding sites, they are costly, time-consuming, and unsuitable for proteins with unknown binding

partners. With the explosive growth of proteins in sequence databases (*UniProt Consortium, 2023*; *Sayers et al., 2023*), developing effective and efficient computational methods to recognize potential binding regions from sequences in a large-scale manner is imperative to fill the gap between genome and phenome.

A conventional way to predict binding interfaces is comparative modeling, which employs alignment algorithms to transfer known binding residues from similar templates to the query proteins (*Zhang et al., 2010*; *Yang et al., 2013*; *Esmaielbeiki et al., 2016*). Nevertheless, this strategy will be seriously restricted when no high-quality template exists. Therefore, methods based on machine learning and deep learning have prevailed in recent years, which can be divided into sequence-based and structure-based approaches according to their used protein information. Sequence-based methods leverage machine learning classifiers (e.g. support vector machine) to learn local binding characteristics from sequence contexts in a sliding window (*Yan and Kurgan, 2017*; *Taherzadeh et al., 2016a*; *Zhao et al., 2018*; *Yu et al., 2013*), or employ deep learning models like transformer *Vaswani et al., 2017* to capture global dependencies (*Wang et al., 2022*; *Yuan et al., 2022c*). Despite requiring only readily available protein sequences, these predictors are of limited accuracy due to the lack of tertiary structure information. On the other hand, experimental structure-based approaches are often more effective. Earlier methods encode protein structures as 2D images (*Xia et al., 2020*) or 3D voxels (*Jiménez et al., 2017*), which are processed via grid-based convolutional neural networks. Current approaches tend to handle protein structures as graphs composed of surface point clouds (*Gainza et al., 2020*; *Li and Liu, 2023*), atoms (*Tubiana et al., 2022*; *Krapp et al., 2023b*) or residues (*Xia et al., 2021*; *Yuan et al., 2021*), and adopt geodesic convolution or graph neural networks (GNNs) to learn the binding-relevant spatial patterns. Unfortunately, the expressive capacities of most methods remain restricted, as the geometry of the structure is not yet fully explored. More importantly, both present sequence- and structure-based predictors are hampered for high-throughput practices at the genome scale for two reasons. Firstly, the dependency on evolutionary profiles from multiple sequence alignments (MSA) for most methods leads to high computational expense and occasionally subpar performance for shallow sequence alignments. Secondly, albeit powerful when native structures are available, structure-based methods will exhibit performance declines for unbound or predicted structures, probably owing to their sensitivity towards details and errors in the structures.

Our previous work (*Yuan et al., 2022b*) has validated the feasibility of exploiting predicted structures from AlphaFold2 (*Jumper et al., 2021*) for training to enhance the model's robustness, yet the computationally intensive structure prediction pipeline still restrains its application to novel sequences absent from the AlphaFold Protein Structure Database (*Tunyasuvunakool et al., 2021*). Since protein sequence can be regarded as a language in biology, unsupervised pre-training with language models has recently been applied to protein sequence representation learning and has displayed competitive or better results than manually engineered evolutionary features in different downstream tasks (*Yuan et al., 2022c*; *Rives et al., 2021*; *Elnaggar et al., 2022*; *Unsal et al., 2022*; *Yuan et al., 2023c*). Based on this, ESMFold (*Lin et al., 2023*) replaces evolutionary information from MSA with a large-scale pre-trained protein language model to directly infer atomic-level protein structure from single sequence. This results in an order-of-magnitude acceleration of prediction while maintaining accuracy nearly as alignment-based methods including AlphaFold2. Therefore, it is promising to develop a fast and accurate model tailored for large-scale sequence-based binding site prediction based on the recent advances in protein representation learning and structure prediction with language models.

To facilitate protein structure modeling, geometric deep learning techniques have recently flourished in protein docking (*Stärk et al., 2022*), protein structure pre-training (*Zhang et al., 2022b*), protein design (*Dauparas et al., 2022*; *Gao et al., 2022*), and binding site prediction (*Gainza et al., 2020*; *Li and Liu, 2023*; *Tubiana et al., 2022*; *Krapp et al., 2023b*), since they can better manipulate data with no natural grid-like topology than 3D convolutional neural networks. Among these, current geometric binding site predictors are mostly built on surface point clouds (*Gainza et al., 2020*; *Li and Liu, 2023*) or atom graphs (*Tubiana et al., 2022*; *Krapp et al., 2023b*). However, although the binding interface is mainly comprised of surface atoms, the global structure of the protein in general influences how the interface is formulated and how the binding partner is interacted with, which should be modeled. Besides, the calculation of protein surfaces and mapping of their properties are usually time-consuming, while methods based on full atom graphs are memory-consuming and thus difficult to process long sequences. Consequently, designing a geometry-aware message passing mechanism

on residue graphs to synergistically integrate sequence and structure information is potentially more suitable for the binding site prediction task.

In this study, we propose GPSite (Geometry-aware Protein binding Site predictor), a fast, accurate, and versatile network for concurrently predicting binding residues of 10 types of biologically relevant molecules including DNA, RNA, peptide, protein, ATP, HEM, and metal ions ($Zn^{2+}$, $Ca^{2+}$, $Mg^{2+}$, $Mn^{2+}$) in a multi-task framework. GPSite was trained on informative sequence embeddings and predicted structures generated by pre-trained protein language models, and thus does not rely on MSA or native structures. To better capture the high-level bio-physicochemical characteristics in the predicted structures, a comprehensive geometric featurizer along with an edge-enhanced graph neural network is designed to extract the residual and relational geometric contexts in an end-to-end manner. Experiments demonstrate that GPSite substantially surpasses state-of-the-art sequence-based and structure-based approaches on various benchmark datasets, even under conditions where the predicted structures are of lower quality. GPSite runs fast enough to process genome-scale sequence databases such as the entire Swiss-Prot (*UniProt Consortium, 2023*), allowing for rapid binding residue annotations for over 568,000 sequences. Further analyses indicate that such annotations can promote discoveries in associations of binding sites with molecular functions, biological processes, and genetic variants. Besides the standalone code, we also provide the GPSite webserver and annotation database at https://bio-web1.nscc-gz.cn/app/GPSite.

## Results

### The geometry-aware protein binding site predictor (GPSite)

GPSite is a geometry-aware versatile protein binding site predictor that can fast and accurately identify binding residues of DNA, RNA, peptide, protein, ATP, HEM, and metal ions ($Zn^{2+}$, $Ca^{2+}$, $Mg^{2+}$, $Mn^{2+}$) from protein sequences. As shown in *Figure 1* and detailed in Methods, an input protein sequence is fed to the pre-trained language model ProtTrans (*Elnaggar et al., 2022*) and the folding model ESMFold to generate informative sequence embedding and predicted structure, respectively. From the predicted structure, the coordinates of the N, $C_\alpha$, C, and O atoms as well as the centroid of the heavy sidechain atoms are gathered, and DSSP (*Kabsch and Sander, 1983*) is employed to calculate the relative solvent accessibility and secondary structure for each residue. Then, a protein radius graph is built where residues constitute the nodes and adjacent nodes are connected by edges. In addition to the residue features by ProtTrans and DSSP, an end-to-end geometric featurizer is designed to construct a local coordinate system for each residue and extract geometric features capturing the arrangements of backbone and sidechain atoms in or between residues. Specifically, the geometric node features consist of intra-residue distances between two atoms (including the sidechain centroid), relative directions of other inner atoms or sidechain centroid to $C_\alpha$, as well as the bond and torsion angles. Similarly, the geometric edge features comprise inter-residue distances between atoms from the two neighboring residues, relative directions of all atoms in the adjacent residue to $C_\alpha$ of the central residue, and spatial orientation between the two reference frames of the neighboring nodes.

To learn the residue representations by considering multi-scale interactions in node, edge, and global context levels, we design a GNN with message passing, edge update and global node update acting on this geometric-aware attributed graph. The message passing layer adopts the multi-head attention in transformer enhanced by edge features to aggregate information from the local neighborhood and update the central node. Then the features of an edge are updated using its connecting nodes. Finally, a gated attention is applied to update the node states using the global context information. Benefiting from the above pipeline, GPSite is invariant to rotation and translation. Besides, GPSite leverages the multi-task learning strategy, where the GNN is shared among different ligands to capture the common binding-relevant characteristics, followed by 10 ligand-specific multilayer perceptrons (MLPs) to mine the binding patterns of particular molecules. This framework also reduces the time for inference of multiple ligands by the concurrent prediction fashion. In conclusion, GPSite is distinguished from the previous approaches in four key aspects. First, profiting from the effectiveness and low computational cost of ProtTrans and ESMFold, GPSite is liberated from the reliance on MSA and native structures, thus enabling genome-wide binding site prediction. Second, unlike methods that only explore the $C_\alpha$ models of proteins (*Xia et al., 2021*; *Ingraham et al., 2019*), GPSite exploits a comprehensive geometric featurizer to fully refine

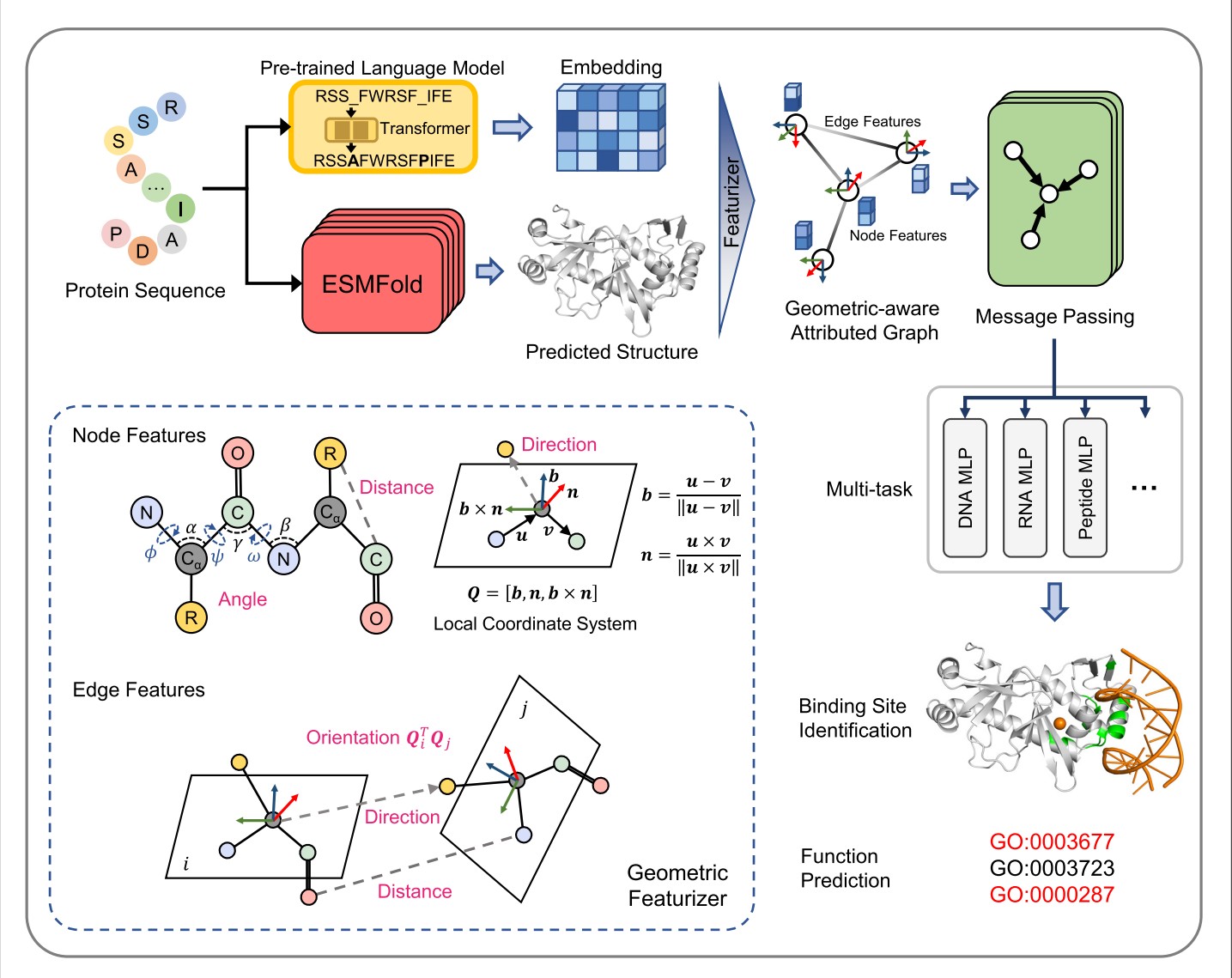

**Figure 1.** The overview of GPSite. The protein sequence is input to the pre-trained language model ProtTrans and the folding model ESMFold to generate the sequence embedding and predicted structure, respectively. According to the structure, a protein radius graph is constructed where residues constitute the nodes and adjacent nodes are connected by edges. In addition to the pre-computed residue features of ProtTrans embedding and DSSP structural properties, a comprehensive, end-to-end geometric featurizer is employed to extract the geometric node features including distance, direction and angle, as well as geometric edge features between residues including distance, direction and orientation. Here, the R group denotes the centroid of the heavy sidechain atoms. The resulting geometric-aware attributed graph is input to a shared GNN to perform edge-enhanced message passing for capturing the common binding-relevant characteristics among different molecules. Finally, 10 ligand-specific MLPs are adopted to learn the binding patterns of particular molecules in a multi-task manner. Examples of the applications of GPSite include binding site identification and protein-level Gene Ontology (GO; *Ashburner et al., 2000*) function prediction.

knowledge in the backbone and sidechain atoms. Third, the employed message propagation on residue graphs is global structure-aware and time-efficient compared to the methods based on surface point clouds (*Gainza et al., 2020*; *Li and Liu, 2023*), and memory-efficient unlike methods based on full atom graphs (*Tubiana et al., 2022*; *Krapp et al., 2023b*). Residue-based message passing is also less sensitive towards errors in the predicted structures. Last but not least, instead of predicting binding sites for a single molecule type or learning binding patterns separately for different molecules, GPSite applies multi-task learning to better model the latent relationships among different binding partners.

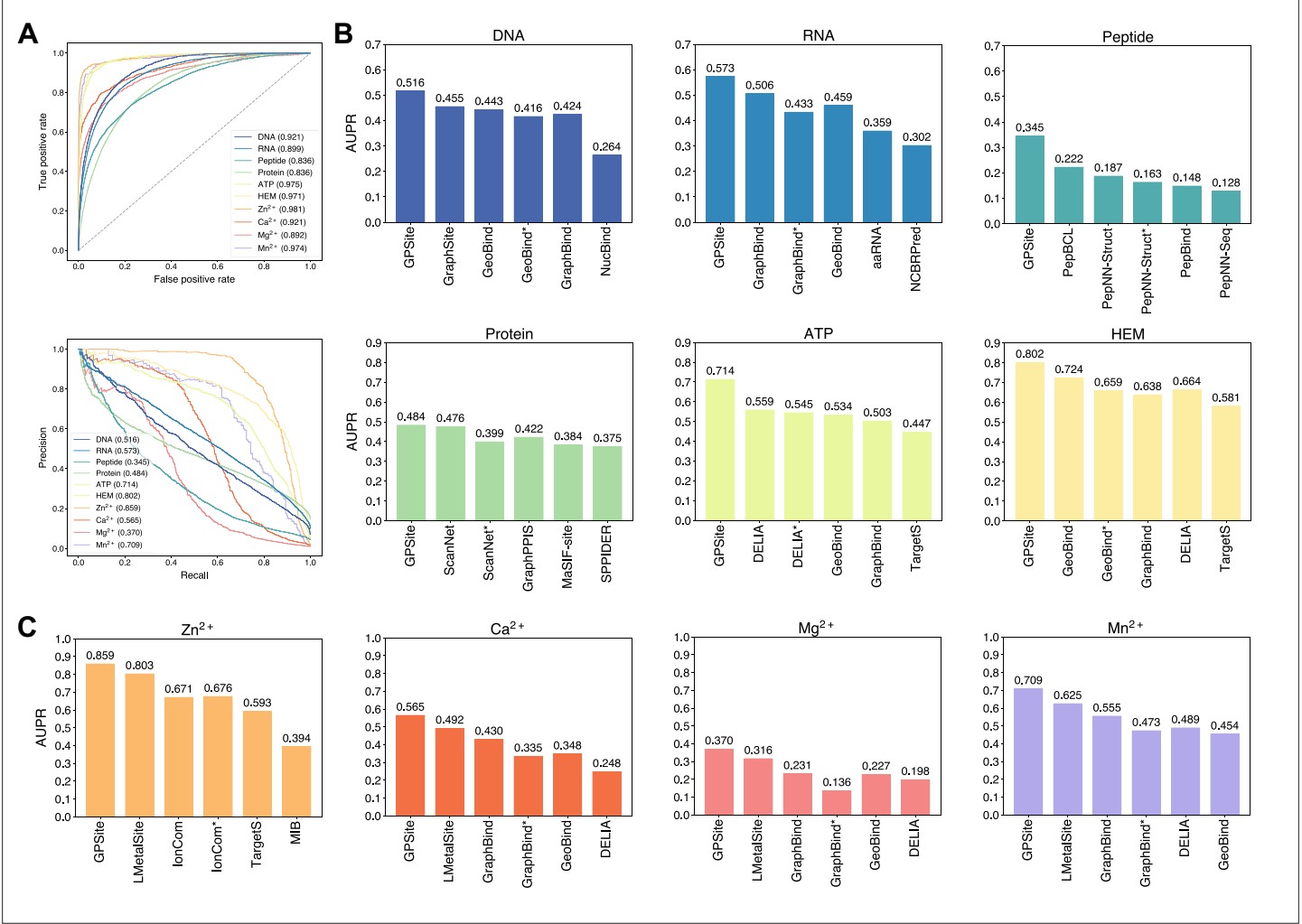

**Figure 2.** The performance of GPSite and the state-of-the-art methods. (**A**) The ROC and precision-recall curves of GPSite on the 10 binding site test sets. The numbers in the legends are areas under the curves. (**B–C**) The AUPR values of the top-performing methods in each test set. The methods marked with * denote evaluations using the ESMFold-predicted structures as input.

## GPSite outperforms state-of-the-art methods

We collected 10 binding site benchmark datasets for the 10 ligands from Protein Data Bank (PDB; *Berman et al., 2000*) as detailed in Methods, which were combined to train and evaluate GPSite using the five-fold cross-validation and independent test sets. As shown in *Appendix 2—table 2*, GPSite obtains average area under the receiver operating characteristic curve (AUC) values over the 10 ligand types of 0.918 and 0.921; as well as average area under the precision-recall curve (AUPR) values of 0.603 and 0.594 on the cross-validation and independent tests, respectively. The consistent performance on the validation and test sets indicates the robustness of our model. In *Figure 2A*, the receiver operating characteristic (ROC) curves and the precision-recall curves on the 10 test sets are plotted to overview the performance of GPSite for different ligands.

To demonstrate the effectiveness of our method, we compared GPSite with 9 sequence-based predictors including DRNApred (*Yan and Kurgan, 2017*), NCBRPred (*Zhang et al., 2021*), SVMnuc (*Su et al., 2019*), GraphSite (*Yuan et al., 2022b*), PepBind (*Zhao et al., 2018*), PepNN-Seq (*Abdin et al., 2022*), PepBCL (*Wang et al., 2022*), TargetS (*Yu et al., 2013*), and LMetalSite (*Yuan et al., 2022c*), as well as 15 structure-based predictors including NucBind (*Su et al., 2019*), COACH-D (*Wu et al., 2018*), GraphBind (*Xia et al., 2021*), GeoBind (*Li and Liu, 2023*), aaRNA (*Li et al., 2014*), PepNN-Struct (*Abdin et al., 2022*), DeepPPISP (*Zeng et al., 2020*), SPPIDER (*Porollo and Meller, 2007*), MaSIF-site (*Gainza et al., 2020*), GraphPPIS (*Yuan et al., 2021*), ScanNet (*Tubiana et al.,*

*2022*), PeSTo (*Krapp et al., 2023b*), DELIA (*Xia et al., 2020*), MIB (*Lin et al., 2016*), and IonCom (*Hu et al., 2016*) (see Brief introductions to the competitive methods for more details). *Figure 2B and C* show the results of the top-performing predictors in the test sets, where GPSite surpasses all other sequence-based and even experimental structure-based methods in AUPR for more than 16.5%, 13.2%, 55.4%, 1.7%, 27.7%, 10.8%, 7.0%, 14.8%, 17.1%, and 13.4% in the DNA, RNA, peptide, protein, ATP, HEM, $Zn^{2+}$, $Ca^{2+}$, $Mg^{2+}$, and $Mn^{2+}$ binding site test sets, respectively. The results of more contending methods and criteria (e.g. AUC and F1-score) are tabulated in *Appendix 2—table 3*. Given the substantial overlap between our protein-binding site test set and the training set of PeSTo, we conducted separate training and comparison using the datasets of PeSTo, where GPSite still demonstrates a remarkable improvement over PeSTo (see Performance comparison between GPSite and PeSTo). Moreover, GPSite is computationally efficient, achieving comparable or faster prediction speed compared to other top-performing methods (*Appendix 3—figure 1*).

Although trained on predicted protein structures, GPSite can also adopt native structures as input for prediction whenever applicable. By doing this, extra performance boosts can be gained with average AUPR increase of 7.8% (*Appendix 3—figure 2*). However, experimental structures are not always available in real-world scenarios, such as genome-scale sequence databases. To this end, for the best experimental structure-based method (measured by AUPR) in each test set, we also investigated the impact on performance when using ESMFold-predicted structures as input. As expected, the performance of these methods mostly decreases substantially utilizing predicted structures for testing, because they were trained with high-quality native structures. For example, the AUPR of GraphBind for predicting RNA-binding sites decreases from 0.506 to 0.433, compared to the AUPR of 0.573 by GPSite. Similarly, the AUPR of ScanNet drops from 0.476 to 0.399, compared to the AUPR of 0.484 by GPSite for predicting protein-binding sites. Therefore, in the practical situations where experimental structures are unavailable, the superiority of our method will be further reflected.

## GPSite is robust for low-quality predicted structures

Since GPSite is built on ESMFold, it is necessary to examine the quality of the predicted structures and its impact on the model performance. *Figure 3B* and *Appendix 3—figure 3* show the distributions of TM-scores between native and predicted structures calculated by US-align (*Zhang et al., 2022a*) in the 10 benchmark datasets, where most proteins are accurately predicted with TM-score >0.7 (see also *Appendix 2—table 5*). Overall, ESMFold achieves median TM-scores of 0.89, 0.76, 0.93, 0.93, 0.94, 0.94, 0.93, 0.94, 0.95, and 0.96 for the DNA, RNA, peptide, protein, ATP, HEM, $Zn^{2+}$, $Ca^{2+}$, $Mg^{2+}$, and $Mn^{2+}$ datasets, respectively (*Appendix 2—table 6*). We next explored whether GPSite can maintain its performance on low-quality predicted structures. *Figure 3A* presents the performance of GPSite on ESMFold-predicted structures with TM-score >0.7 or ≤0.7, and the comparisons with the leading structure-based methods in the test sets of DNA, RNA, and peptide. Reasonably, compared to the well-predicted proteins, the performance of GPSite is inferior on the subsets of proteins with TM-score ≤0.7. Nevertheless, GPSite continues to outshine the most advanced structure-based methods input with ESMFold-predicted structures or even experimental structures. Similar trends are also observed for the rest of the ligands in *Appendix 3—figure 4*. Given the infrequency of low-quality predicted structures except for the RNA test set, we took a closer inspection of the 104 proteins with predicted structures of TM-score <0.5 in the RNA test set. In this subset, GraphBind achieves AUPR values of 0.455 and 0.376 using native and predicted structures, respectively, compared to the AUPR of 0.516 by GPSite. As shown in *Figure 3C* with lines fit to the per-protein TM-score and AUPR using linear regression, GPSite consistently outperforms GraphBind using predicted structures regardless of the prediction quality of ESMFold, and is only surpassed by GraphBind input with native structures on proteins of extremely low quality (TM-score <0.3). An example is presented in Case study for the ribosome biogenesis protein ERB1 for illustration. To sum up, ESMFold could produce high-quality structures in most cases, and even for the low-quality predicted structures, GPSite is robust enough to generate reliable predictions better than current state-of-the-art structure-based methods.

We finally illustrate a potential reason for the robustness of GPSite by an example from the test set where GPSite is able to discern among various interfaces even though the structure is not perfectly predicted. *Figure 3D* shows the structure of the human glucocorticoid receptor (GR), a transcription factor that binds DNA and assembles a coactivator peptide to regulate gene transcription (PDB: 7PRW, chain A). The DNA-binding domain of GR also consists of two C4-type zinc fingers to bind

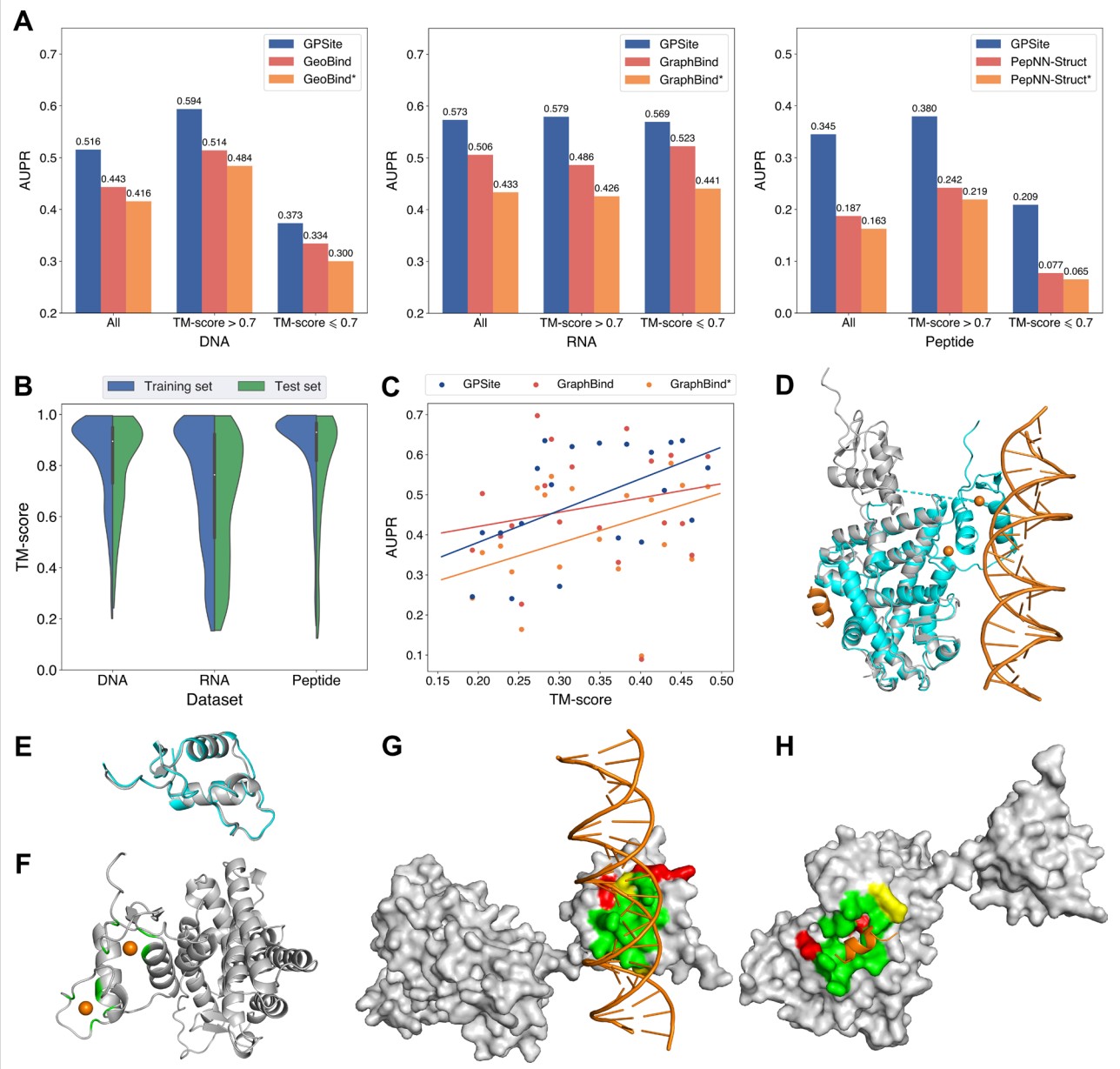

**Figure 3.** The performance of GPSite on low-quality predicted structures. (**A**) The performance of GPSite on structures of different qualities, and the comparisons with the best experimental structure-based methods in the test sets of DNA, RNA, and peptide. The experimental structure-based methods input with ESMFold-predicted structures are marked with *. (**B**) Distributions of the TM-scores between native and predicted structures in the DNA, RNA and peptide datasets. (**C**) The correlations between the prediction quality of ESMFold and the performance of GPSite and GraphBind on the RNA-binding site test set when TM-score <0.5. The scatters denote the average TM-score and AUPR for each bin after sorting the proteins according to the TM-scores and evenly dividing them into 20 discrete bins. The lines are fit to the original data (without binning) using linear regression. (**D**) The glucocorticoid receptor (GR) in complex with DNA, a coactivator peptide, and $Zn^{2+}$ ions (PDB: 7PRW). The ESMFold-predicted protein structure (gray) is superimposed to the native structure (cyan) using US-align (TM-score=0.72). The ligands are colored in orange. (**E**) Superposition of the native (cyan) and predicted (gray) DNA-binding domains of GR (TM-score=0.96). (**F–H**) The $Zn^{2+}$, DNA and peptide binding site predictions by GPSite for the predicted GR structure in cartoon or surface view. True positives, false positives and false negatives are colored in green, red and yellow, respectively. The ligands in orange were subsequently added based on the native complex structure to show the quality of the predictions by GPSite.

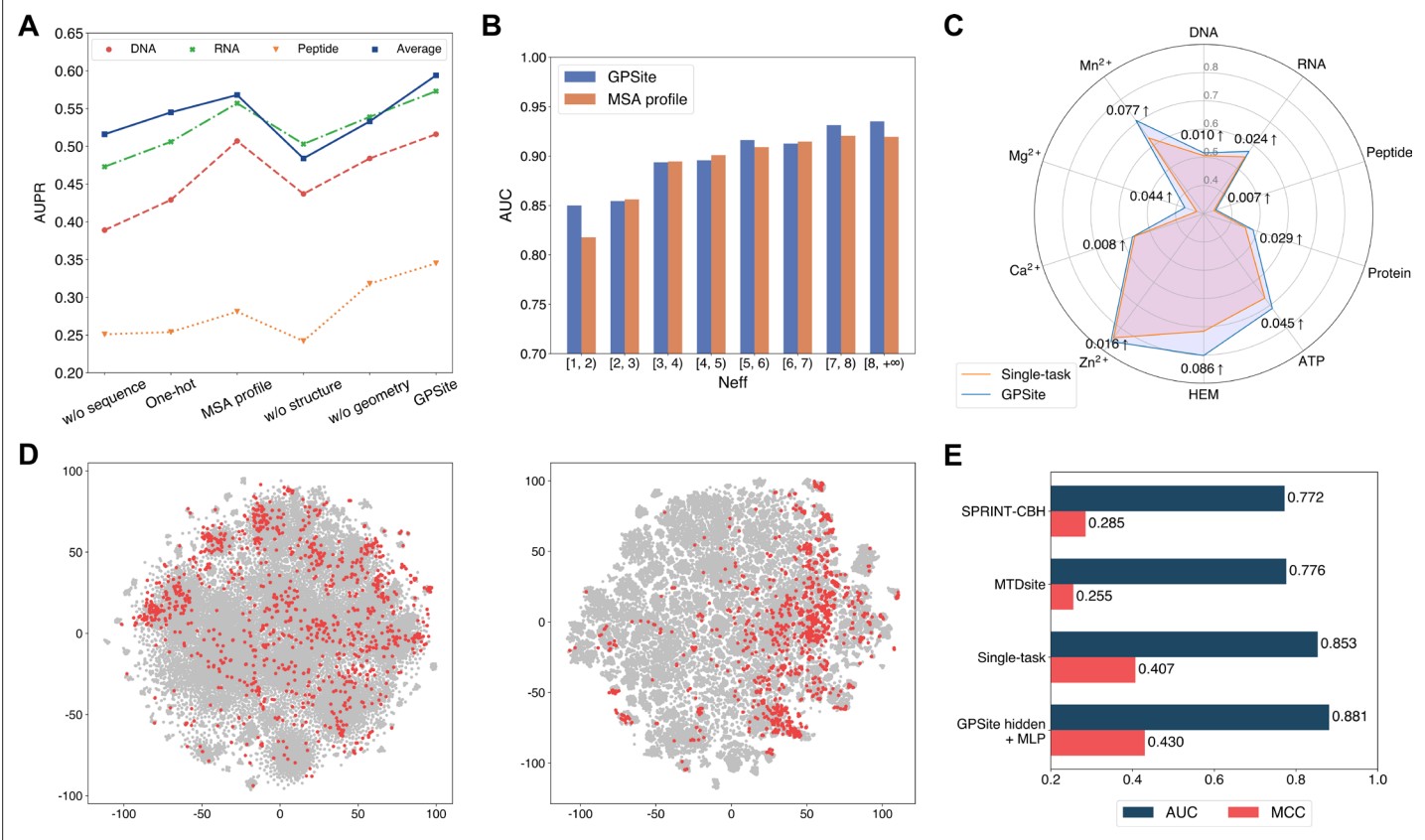

**Figure 4.** The effects of protein features and model designs. (**A**) Ablation studies on sequence and structure information in the DNA, RNA, and peptide test sets. The average performance of the 10 test sets is also shown. (**B**) Performance comparison between GPSite and the baseline model using MSA profile for proteins with different Neff values in the combined test set of the 10 ligands. (**C**) Performance boosts in AUPR using GPSite compared to the single-task baseline. (**D**) Visualization of the distributions of residues encoded by raw feature vectors (left) or hidden embedding vectors from the pre-trained shared network in GPSite (right) for the unseen carbohydrate-binding site dataset using t-SNE. The binding and non-binding residues are colored in red and gray, respectively. (**E**) The performance when using the hidden embeddings from GPSite as input features to train an MLP for carbohydrate-binding site prediction, and its comparisons with other methods.

$Zn^{2+}$ ions. Although the structure of this protein is not perfectly predicted (TM-score=0.72), the local structures of the binding domains of peptide and DNA are actually predicted accurately as viewed by the superpositions of the native and predicted structures in *Figure 3D and E*. Therefore, GPSite can correctly predict all $Zn^{2+}$ binding sites and precisely identify the binding sites of DNA and peptide with AUPR values of 0.949 and 0.924, respectively (*Figure 3F, G and H*).

## The effects of protein features and model designs

To reveal the roles of distinct protein features and model designs in GPSite, we conducted comprehensive ablation studies. As shown in *Figure 4A* and *Appendix 2—table 7*, when removing the Prot-Trans embeddings from GPSite, the model yields inadequate performance (average AUPR of 0.516 among the 10 test sets) due to the complete neglect of the sequence information in proteins. The introduction of one-hot sequence encodings or MSA profile (elaborated in Generation of the evolutionary features from MSA) partially restores the performance to average AUPR of 0.545 or 0.568, respectively. Nevertheless, the utilization of language model representations in GPSite attains the highest average AUPR of 0.594. To further understand the advantages of ProtTrans over the evolutionary features from MSA, we compared their performance against the number of effective homologous sequences (Neff) of the proteins from the combined test set of the 10 ligands. Neff is an HHblits (*Remmert et al., 2011*) parameter quantifying the effective size of homologous sequence cluster. As evidenced in *Figure 4B* and *Appendix 2—table 8*, ProtTrans consistently obtains competitive or superior performance compared to the MSA profile. Notably, for the target proteins with few

homologous sequences (Neff <2), ProtTrans surpasses MSA profile significantly with an improvement of 3.9% on AUC (p-value = $4.3 \times 10^{-8}$). On the other hand, removing the structure information (implemented by a transformer model solely input with ProtTrans sequence features) obtains the worst performance with an average AUPR of 0.484 (*Figure 4A*). This observation indicates that the knowledge of protein structure may be more critical than sequence information in binding site prediction tasks. Additionally, the removal of the geometric featurizer within GPSite also causes a substantial decline in performance (average AUPR from 0.594 to 0.533), attesting to the significance of GPSite's perception of protein geometry. We also assessed the benefit of training with predicted instead of native structures, which brings an average AUPR increase of 4.2% as detailed in The effect of training with predicted structures.

We next elucidate the benefits of the multi-task framework in GPSite by comparing it with a baseline approach in which a model is trained and evaluated for each dataset separately. As depicted in *Figure 4C* and *Appendix 2—table 7*, GPSite consistently outperforms the single-task baseline, especially for the ligands with limited training data. For instance, directly fitting a model on the HEM training set with only 176 proteins reaches an AUPR of 0.716 for the test set. Alternatively, combining datasets of diverse ligands in a multi-task framework brings an AUPR increase of 0.086 for HEM. This suggests that multi-task learning can compensate for the scarcity of training data by leveraging a shared network trained on a larger dataset encompassing different types of ligand-binding proteins that potentially share similar binding patterns. We also conducted cross-type evaluations to investigate the specificity of the ligand-specific MLPs and the inherent similarities among different ligands in The cross-type performance of the multi-task network in GPSite.

Residues that are conserved during evolution, exposed to solvent, or inside a pocket-shaped domain are inclined to participate in ligand binding. During the preceding multi-task training process, the shared network in GPSite should have learned to capture such common binding mechanisms. Here we show how GPSite can be easily extended to the binding site prediction for other unseen ligands by adopting the pre-trained shared network as a feature extractor. We considered a carbohydrate-binding site dataset from *Sun et al., 2022* which contains 100 proteins for training and 49 for testing. We first visualized the distributions of residues in this dataset using t-SNE (*van der Maaten and Hinton, 2008*), where the residues are encoded by raw feature vectors encompassing ProtTrans embeddings and DSSP structural properties, or latent embedding vectors from the shared network of GPSite trained on the 10 molecule types previously. As shown in *Figure 4D*, the binding and non-binding residues overlap and are indistinguishable when encoded by raw feature vectors. On the contrary, the latent representations from GPSite effectively improve the discriminability between the binding and non-binding residues. Employing these informative hidden embeddings as input features to train a simple MLP exhibits remarkable performance with an AUC of 0.881 (*Figure 4E*), higher than that of training a single-task version of GPSite from scratch (AUC of 0.853) or other state-of-the-art methods such as MTDsite (*Sun et al., 2022*) and SPRINT-CBH (*Taherzadeh et al., 2016b*). These results highlight the effectiveness of multi-task learning and the scalability of GPSite to unseen ligands.

## Large-scale binding site annotation for Swiss-Prot

In light of the efficiency and effectiveness of GPSite, we sought to annotate and analyze the potential binding interfaces of various kinds for the entire Swiss-Prot database. For this task, we applied ESMFold to predict the structures of 568,326 sequences in Swiss-Prot, which required approximately 8.5 days as described in Methods. Typically, it takes 16 s to predict the structure of a protein with 500 residues, or 100 s for 1000 residues (*Appendix 3—figure 6*). The feature extraction and GPSite inference procedures overall cost about 5 hr. All ESMFold-predicted structures accompanied by the binding site annotations for Swiss-Prot are freely available in our user-friendly GPSiteDB database (https://bio-web1.nscc-gz.cn/database/GPSiteDB). *Appendix 3—figure 7* further illustrates the distributions of the protein length and the predicted TM-score (pTM) estimated by ESMFold for the Swiss-Prot sequences, where most proteins are no longer than 500 residues and predicted with high confidence (pTM >0.8). For the subsequent downstream analyses, we only considered the predicted structures with pTM >0.7, resulting in a total of 370,140 structures.

Exploiting the residue-level binding site annotations, we could readily extend GPSite to discriminate between binding and non-binding proteins of various ligands. Specifically, a protein-level binding score indicating the overall binding propensity to a specific ligand can be generated by averaging the

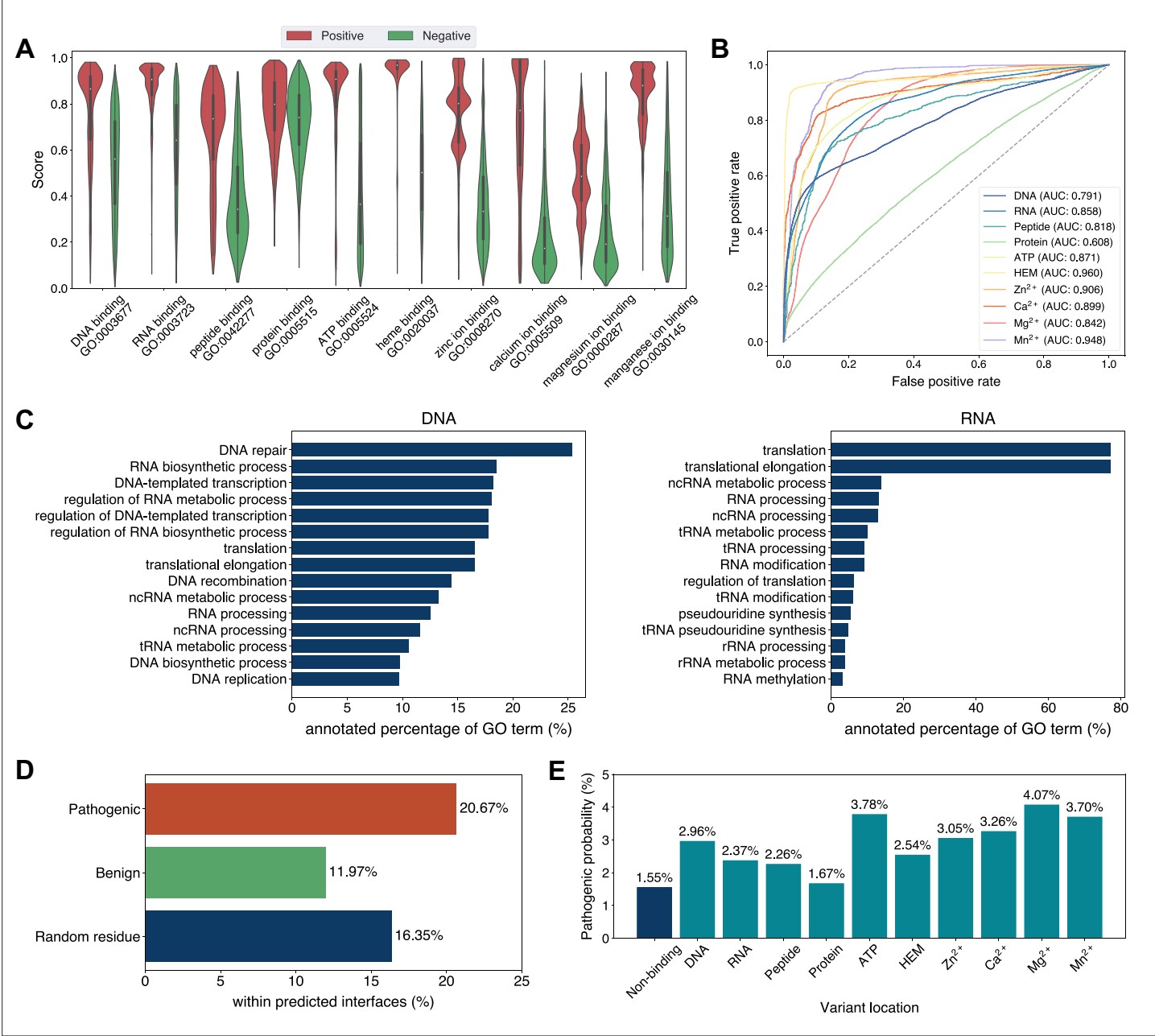

**Figure 5.** Analyses of Swiss-Prot based on the binding site annotations by GPSite. (**A**) The distributions of the binding scores assigned by GPSite for proteins with or without certain ligand-binding molecular function in GO. (**B**) The ROC curves when using the GPSite binding scores to distinguish between binding and non-binding proteins of various ligands. (**C**) The percentage of proteins predicted as binding to DNA and RNA by GPSite to be annotated with certain biological process in Swiss-Prot. Only the specific biological process terms with depth ≥8 in the GO directed acyclic graph are considered, among which the top 15 terms with the highest percentages are displayed. (**D**) The percentage of surface pathogenic or benign natural variant sites within GPSite-predicted interfaces. The baseline is the probability of a random surface residue being annotated as an interface residue. (**E**) The pathogenic probabilities of variants located in non-binding sites or different types of binding sites predicted by GPSite.

top *k* predicted scores among all residues. Empirically, we set *k* to 5 for metal ions and 10 for other ligands, considering the distributions of the numbers of binding residues per sequence observed in the training set. As depicted in *Figure 5A*, the GPSite binding scores for proteins with the corresponding ligand-binding molecular functions are significantly higher than those without such annotations in Swiss-Prot (p-value $<10^{-165}$ for all ligands according to Mann–Whitney *U* test; *Mann and Whitney, 1947*). The accuracy of the GPSite protein-level binding scores is further validated by the

ROC curves in *Figure 5B*, where GPSite achieves satisfactory AUC values for all ligands except protein (AUC of 0.608). This may be ascribed to the fact that protein-protein interactions are ubiquitous in living organisms while the Swiss-Prot function annotations are incomplete (see GPSite is effective for completing the function annotations in Swiss-Prot). Moreover, we attempted to gather the top 20,000 proteins with the highest GPSite binding scores for each ligand to expand the binding function annotations in Swiss-Prot. We could immediately notice that the GPSite-predicted binding proteins are involved in biological processes consistent with existing knowledge as shown in *Figure 5C* and *Appendix 3—figure 8*. For instance, the DNA-binding proteins predicted by GPSite are prone to participate in DNA repair, DNA-templated transcription, DNA recombination and replication, while the RNA-binding proteins are inclined to perform translation and RNA modification.

Capitalizing on the predicted structures and annotations within GPSiteDB, cell biologists are empowered to easily locate the genetic variants and assess their potential disruptions in protein-ligand interactions and pathogenicity. This facilitates the establishment of rational working hypotheses to propel therapeutic development in a more informed manner. Here we conduct analyses on the associations between binding sites and genetic variants for the human proteome as an example. Notably, 20.67% of the pathogenic variant sites on the surfaces of the predicted structures fall in the GPSite-predicted interfaces, higher than the benign variants (11.97%) or the random baseline (16.35%) as described in *Figure 5D*. Consistent trend is observed in *Appendix 3—figure 9* when considering variants in the entire structure (rather than solely on surface). Besides, we investigated the pathogenic probabilities of variants in different locations in *Figure 5E*. As expected, the pathogenicity of variants located in the predicted binding sites is higher than those in the non-binding sites. Interestingly, our analysis uncovered that the pathogenic probabilities of variants in the predicted binding sites of ATP and metal ions surpass those of other ligands. One possible reason is that the binding interfaces of ATP and metal ions typically comprise small pockets formed by a limited number of residues. Consequently, variants affecting even a single residue within such pockets may exert a substantial influence on the overall pocket functionality and lead to diseases.

## Discussion

In this study, we present GPSite to accurately and efficiently predict protein binding sites of diverse biologically relevant molecules including DNA, RNA, peptide, protein, ATP, HEM, and metal ions. By leveraging the informative sequence embeddings and predicted structures from pre-trained language models, GPSite is liberated from the reliance on MSA or experimental protein structures. To encapsulate the high-level bio-physicochemical characteristics in the predicted structures, GPSite incorporates a comprehensive geometric featurizer and an edge-enhanced graph neural network to refine both residual and relational geometric contexts in an end-to-end manner. GPSite also stands out from the previous approaches by applying a multi-task framework to effectively model the intrinsic relationships among different binding partners. Results across various benchmark datasets indicate that GPSite substantially outperforms state-of-the-art sequence-based and structure-based methods, even under conditions where the predicted structures are of lower quality. Finally, we demonstrate GPSite's scalability to genome-scale sequence databases by annotating binding sites for over 568,000 sequences in Swiss-Prot within 9 days. Further analyses suggest that these annotations are not only accordant with existing knowledge but also capable of facilitating discoveries of unexplored biology in protein function and genetic variant.

Despite the noteworthy advancements achieved by GPSite, there remains scope for further improvements. GPSite may be improved by pre-training (*Zhang et al., 2022b*) on the abundant predicted structures in ESM Metagenomic Atlas (*Lin et al., 2023*), and then fine-tuning on binding site datasets. Besides, the hidden embeddings from ESMFold may also serve as informative protein representations. Additional opportunities for upgrade exist within the network architecture. For example, a variational Expectation-Maximization framework (*Zhao et al., 2022*) can be adopted to handle the hierarchical atom-to-residue graph structure inherent in proteins. Meta-learning (*Finn et al., 2017*) could also be explored in this multi-task scenario, which allows fast adaptation to unseen tasks with limited labels.

As the gap between unannotated and annotated sequences is expanding at an unparalleled rate, GPSite serves as a reliable, efficient, versatile and user-friendly tool for unraveling the extensive and dynamic landscape of protein-ligand interactions. By harnessing the capabilities of GPSite, researchers

can readily uncover fresh biological functions of proteins, gain valuable insights into the underlying pathogenic mechanisms of gene mutations, or design novel drugs targeting specific binding pockets.

## Methods

### Benchmark datasets

The benchmark datasets for evaluating binding site predictions of DNA, RNA, peptide, ATP, and HEM are constructed from BioLiP (*Zhang et al., 2024*), a database of biologically relevant protein-ligand complexes primarily from PDB. For each ligand, we collected the corresponding binding proteins with resolutions of ≤3.0 Å and lengths of 50–1500 from BioLiP released on 29 March 2023. A binding residue is defined if the smallest atomic distance between the target residue and the ligand is <0.5 Å plus the sum of the Van der Waal's radius of the two nearest atoms. We combined the binding site annotations of identical sequences and then removed redundant proteins sharing sequence identity >25% over 30% alignment coverage using CD-HIT (*Fu et al., 2012*). Finally, each benchmark dataset was split into a training set with proteins released before 1 January 2021, as well as an independent test set with proteins released from 1 January 2021 to 29 March 2023. Besides, the benchmark dataset of protein-protein binding sites is directly from *Yuan et al., 2021*, which contains non-redundant transient heterodimeric protein complexes dated up to May 2021. Surface regions that become solvent inaccessible on complex formation are defined as the ground truth protein-binding sites. The benchmark datasets of metal ion ($Zn^{2+}$, $Ca^{2+}$, $Mg^{2+}$, and $Mn^{2+}$) binding sites are directly from *Yuan et al., 2022c*, which contain non-redundant proteins dated up to December 2021 from BioLiP. Combining all these 10 datasets results in a total of 8441 training sequences and 1838 test sequences. Details of the statistics of these benchmark datasets are given in *Appendix 2—table 1*.

### Structure prediction and preprocessing

We harnessed ESMFold, a fast and accurate end-to-end model to predict protein structures from sequences. ESMFold is based on a language model with 3B parameters pre-trained over sequences in UniRef50 (*Suzek et al., 2007*), and a folding head similar to AlphaFold2 trained on experimental structures from PDB and predicted structures from AlphaFold2. The structure prediction for our whole benchmark datasets (~10,200 sequences, ~300 amino acids on average) cost only ~28 hr on an NVIDIA A100 GPU. For each residue in the predicted structures, we gathered the coordinates of the N, $C_\alpha$, C and O atoms as well as the centroid of the heavy sidechain atoms (denoted as R). In this way, the structure of a protein can be represented by a coordinate matrix $X \in \mathbb{R}^{n \times 5 \times 3}$, with $n$ denoting the number of residues.

### Protein features

GPSite leverages the pre-trained protein language model ProtTrans (version: ProtT5-XL-U50) to generate sequence features efficiently, thus bypassing slow sequence alignments. ProtTrans is a transformer-based auto-encoder named T5 (*Raffel et al., 2020*) pre-trained with the BERT's denoising objective (*Kenton and Toutanova, 2019*), essentially learning to predict the masked amino acids. Concretely, ProtTrans contains 3B parameters, which was first trained on BFD (*Steinegger et al., 2019*) and then fine-tuned on UniRef50. We extracted the output from the last ProtTrans encoder layer as sequence representations, containing a 1024-dimensional vector for each residue. The inference cost of ProtTrans is extremely low, and the embedding extraction process for our whole benchmark datasets can be done within 5 min on an NVIDIA A100 GPU. The feature values in the sequence embeddings are further normalized to scores between 0 and 1 as follows:

$$v_{norm} = \frac{v - v_{min}}{v_{max} - v_{min}} \qquad (1)$$

where $v$ is the original feature value, and $v_{min}$ and $v_{max}$ are the minimum and maximum values of this feature type observed in the training set, respectively. In addition, we also calculated two structural properties from the predicted structures using DSSP: (i) Relative solvent accessibility (RSA), which is the normalized solvent accessible surface area (ASA) by the maximum ASA of the corresponding amino acid type. (ii) One-hot secondary structure profile representing one of the eight secondary structure states.

## The architecture of GPSite

The overall architecture of GPSite is shown in *Figure 1*. First, the protein sequence is input to the pre-trained language model ProtTrans and the folding model ESMFold to generate the sequence embedding and predicted structure, respectively. Second, a protein radius graph is constructed from the structure, where residues constitute the nodes and adjacent nodes (distance between $C_\alpha$<15 Å) are connected by edges. In addition to the pre-computed residue features (ProtTrans embedding and structural properties by DSSP), a comprehensive, end-to-end geometric featurizer is employed to extract the geometric node features including distance, direction and angle, as well as geometric edge features between residues including distance, direction and orientation. Third, the resulting geometric-aware attributed graph is input to a shared GNN with message passing, edge update and global node update, to capture the common binding-relevant characteristics among different molecules. Finally, 10 ligand-specific MLPs are adopted to learn the binding patterns of particular molecules in a multi-task manner.

## The geometric featurizer

GPSite represents the protein as a radius graph derived from the $C_\alpha$ coordinates of the residues, where the radius is equal to 15 Å. An end-to-end featurizer is utilized to act directly on the atomic coordinate matrix $X$ for geometric feature extraction similar to *Gao et al., 2022*, except that we additionally encode the sidechain conformations of the residues. In this representation, a local coordinate system is first defined at each residue based on the relative position of the $C_\alpha$ atom to other backbone atoms. Then, several geometric node and edge features are derived to capture the arrangements of backbone and sidechain atoms in or between residues.

(i) Local coordinate system. We define a local coordinate system $Q_i = [b_i, n_i, b_i \times n_i]$ for residue $i$, where $b_i$ is the negative bisector of the angle formed by the N, $C_\alpha$, and C atoms, and $n_i$ is a unit vector normal to this plane. Formally, we have:

$$u_i = C_{\alpha_i} - N_i, \, v_i = C_i - C_{\alpha_i}, \, b_i = \frac{u_i - v_i}{\|u_i - v_i\|}, n_i = \frac{u_i \times v_i}{\|u_i \times v_i\|} \tag{2}$$

Based on the local coordinate systems, we could construct geometric features that are invariant to rotation and translation for single or pair of residues.

(ii) Geometric node features. GPSite constructs distance, direction and angle features for each residue. Given the coordinates of two atoms $A$ and $B$, the distance feature is computed via $\text{RBF}\left(\|A - B\|\right)$, where $\text{RBF}\left(\cdot\right)$ is a radial basis function. For the intra-residue distance features of node $i$, $A, B \in \{N_i, C_{\alpha_i}, C_i, O_i, R_i\}$ and $A \neq B$. Here, $R$ denotes the centroid of the heavy sidechain atoms. The direction features encoding relative directions of other inner atoms to $C_\alpha$ in residue $i$ are computed via $Q_i^T \frac{A - C_{\alpha_i}}{\|A - C_{\alpha_i}\|}$, where $A \in \{N_i, C_i, O_i, R_i\}$. As shown in *Figure 1*, we also incorporate the sine and cosine values of the bond angles $(\alpha_i, \beta_i, \gamma_i)$ and torsion angles $(\phi_i, \psi_i, \omega_i)$ to consider the backbone geometry.

(iii) Geometric edge features. Similarly, we construct geometric features between neighboring residues including distance, direction and orientation. The inter-residue distance features $\text{RBF}\left(\|A - B\|\right)$ between nodes $i$ and $j$ are computed with atoms $A \in \{N_i, C_{\alpha_i}, C_i, O_i, R_i\}$ and $B \in \{N_j, C_{\alpha_j}, C_j, O_j, R_j\}$. The edge direction features $Q_i^T \frac{A - C_{\alpha_i}}{\|A - C_{\alpha_i}\|}$ consider relative directions of all atoms in residue $j$ to $C_{\alpha_i}$, namely $A \in \{N_j, C_{\alpha_j}, C_j, O_j, R_j\}$. To reflect the relative spatial rotation between the two reference frames of residues $i$ and $j$, the orientation feature $q\left(Q_i^T Q_j\right)$ is employed, where $q\left(\cdot\right)$ is the quaternion encoding function representing 3D rotation matrices as four-element vectors (*Huynh, 2009*).

## The edge-enhanced graph neural network

The above-mentioned attributed graph with features from ProtTrans, DSSP and the geometric featurizer is input to several GNN layers with message passing, edge update and global node update modules, to learn the residue representations by considering multi-scale interactions in node, edge, and global context levels.

(i) Message passing with graph transformer. Since transformer is well-acknowledged as the most powerful network in modeling sequence and graph data (*Yuan et al., 2022c*; *Ingraham et al., 2019*; *Zheng et al., 2020*), we adopt its multi-head attention mechanism while taking the edge features into

account for message passing in graphs. Formally, we denote the hidden feature vectors of node $i$ and edge $j \rightarrow i$ in layer $l$ as $h_i^l$ and $e_{ji}^l$, respectively. Before the first GNN layer, we apply an MLP to project the initial node and edge features to the $d$-dimensional space. To update node $i$, the message passing in layer $l$ is performed as follows:

$$\hat{h}_i^{l+1} = h_i^l + \sum_{j \in N(i) \cup i} \alpha_{ji}^l \left( W_V^l h_j^l + W_E^l e_{ji}^l \right)$$

(3)

where the attention coefficient $\alpha_{ji}^l$ from node $j$ to $i$ is calculated by:

$$\begin{cases} w_{ji}^l = \dfrac{\left( W_Q^l h_i^l \right)^T \left( W_K^l h_j^l + W_E^l e_{ji}^l \right)}{\sqrt{d}} \\ \alpha_{ji}^l = \dfrac{\exp w_{ji}^l}{\sum_{k \in N(i) \cup i} \exp w_{ki}^l} \end{cases}$$

(4)

The learnable weight matrices $W_Q^l$, $W_K^l$ and $W_V^l$ are used to project the node feature vectors into the corresponding query, key and value representations. $W_E^l$ is used to transform the edge features which will be subsequently added to the key and value representations. $N(i)$ denotes the neighbors of node $i$. In practice, we use multi-head attention to linearly project the queries, keys and values multiple times, perform the attention function in parallel and finally concatenate them together.

(ii) Edge update. To improve the model's capability, we update the features of an edge using its connecting nodes:

$$e_{ji}^{l+1} = e_{ji}^l + \text{EdgeMLP} \left( \hat{h}_j^{l+1} \parallel e_{ji}^l \parallel \hat{h}_i^{l+1} \right)$$

(5)

where $\parallel$ denotes the vector concatenation and EdgeMLP is an MLP for edge update.

(iii) Node update with global context attention. While the local node and edge interactions play crucial roles in learning residue representations, the global information is also valuable for improving accuracy. However, global self-attention across the whole protein is computationally intensive. Alternatively, here we learn a global context vector for each protein and use it to apply gated attention for the node representations similar to *Gao et al., 2022*:

$$c^l = \frac{\sum_{k=0}^{n-1} \hat{h}_k^{l+1}}{n}$$

(6)

$$h_i^{l+1} = \hat{h}_i^{l+1} \odot \sigma \left( \text{GateMLP} \left( c^l \right) \right)$$

(7)

where $n$ is the number of residues in a protein, GateMLP is an MLP for gated attention, $\sigma(\cdot)$ is the sigmoid function, and $\odot$ denotes the element-wise product operation.

## Multi-task learning

To better capture the intrinsic similarities of binding patterns among different ligands and enable efficient predictions in a concurrent fashion, GPSite employs a multi-task framework, where the shared edge-enhanced GNN is used to model the common binding-relevant characteristics, followed by 10 ligand-specific MLPs to mine the binding patterns of particular molecules. In the training steps, different types of ligand-binding proteins are input to the same network, and predictions for the 10 types of ligands are yielded. Nonetheless, only predictions with the corresponding known ligand-binding sites are used to calculate loss and perform backpropagation, while the predictions of other ligands without ground truth data are masked. That is, each protein is used to train the shared GNN and the corresponding ligand-specific MLP(s) of its known binding partner(s) without affecting other irrelevant MLP(s).

## Implementation and evaluation

We performed five-fold cross-validation on the training data, where the 10 training sets were mixed and split into five folds randomly, and then each time a model was trained on four folds and evaluated

on the remaining fold. This process was repeated for five times and the average validation performance was used to optimize the hyperparameters of the network. In the test phase, all five trained models from cross-validation were used to make predictions, which were averaged as the final prediction of GPSite. Specifically, we adopted Pytorch 1.13.1 (*Paszke et al., 2019*) to implement GPSite, which contains a four-layer shared GNN with 128 hidden units and four attention heads. The Adam optimizer (*Kingma and Ba, 2014*) with the one-cycle learning rate policy (*Smith and Topin, 2019*) was used for model optimization on the binary cross entropy loss. Within each epoch, we randomly drew 25,000 samples from the training data with replacement to train our model using a batch size of 16. The training process lasted at most 25 epochs and we performed early stopping based on the validation performance, which took ~1.5 hr on an NVIDIA A100 GPU.

Similar to the previous works, we use recall, precision, accuracy, F1-score, Matthews correlation coefficient (MCC), AUC, and AUPR to evaluate the prediction performance, whose detailed definitions are given in Evaluation metrics. AUC and AUPR are independent of thresholds, thus reflecting the overall performance of a model. The other metrics are calculated by converting the predicted binding probabilities to binary predictions with a threshold for each ligand, which is determined by maximizing MCC on the validation sets. We adopted AUPR for hyperparameter selections as it is more sensitive and informative than AUC in imbalanced two-class classification tasks (*Saito and Rehmsmeier, 2015*).

## High-throughput annotation and analysis on Swiss-Prot

We downloaded all the available 569,516 sequences in Swiss-Prot (release: 2023-05-03) and then removed sequences longer than 2700 residues (0.21%) due to the memory limit of GPUs, resulting in a total of 568,326 sequences. Non-standard amino acids in these sequences were also removed. ESMFold was applied to predict the structures from sequences on 16 NVIDIA A100 (80 GB) GPUs, which cost ~8.5 days. The structure preprocessing and feature extraction (by ProtTrans and DSSP) procedures overall cost ~4 hr on the same GPU cluster, and the inference of binding sites using GPSite took ~1 hr.

For the downstream analyses of protein function and variant, we only considered the predicted protein structures with length ≥50 and pTM >0.7 evaluated by ESMFold, consisting of 370,140 structures eventually. Interface residues are defined as residues with predicted binding probabilities higher than the pre-defined thresholds described in implementation and evaluation. Surface residues on the predicted structures are defined as the residues with RSA >5% (*Jones and Thornton, 1997*) computed by DSSP. The annotated GO terms of molecular function and biological process for all sequences were downloaded from UniProt (*UniProt Consortium, 2023*), and we up-propagated the annotations using all types of relationships defined in the hierarchical structure of GO (release: 2023-05-10). The binding proteins of a specific ligand were determined as those annotated with the corresponding ligand-binding molecular function, and the non-binding proteins were randomly sampled to the same number as binding proteins. Concretely, we collected 21680, 42074, 1240, 24108, 74428, 4960, 15030, 2088, 24161, and 4093 binding proteins from Swiss-Prot for DNA, RNA, peptide, protein, ATP, HEM, $Zn^{2+}$, $Ca^{2+}$, $Mg^{2+}$, and $Mn^{2+}$ respectively. The pathogenicity annotations of human protein altering variants were downloaded from UniProt (release: 2023_02), which contain UniProt manually reviewed natural variants, as well as variants imported from other public resources such as Ensembl Variation (*Martin et al., 2023*) and ClinVar (*Landrum et al., 2018*), and the conflicting annotations were removed.

## Acknowledgements

This study has been supported by the National Key Research and Development Program of China (2022YFF1203100) and National Natural Science Foundation of China (T2394502).

## Additional information

### Funding

| Funder | Grant reference number | Author |
|---|---|---|
| National Key Research and Development Program of China | 2022YFF1203100 | Yuedong Yang |
| National Natural Science Foundation of China | T2394502 | Yuedong Yang |

The funders had no role in study design, data collection and interpretation, or the decision to submit the work for publication.

### Author contributions

Qianmu Yuan, Conceptualization, Data curation, Software, Formal analysis, Validation, Investigation, Visualization, Methodology, Writing - original draft; Chong Tian, Software; Yuedong Yang, Conceptualization, Resources, Supervision, Funding acquisition, Project administration, Writing - review and editing

### Author ORCIDs

Qianmu Yuan ![ORCID] http://orcid.org/0000-0001-6098-9103
Yuedong Yang ![ORCID] http://orcid.org/0000-0002-6782-2813

Reviewer #1 (Public review): https://doi.org/10.7554/eLife.93695.3.sa1
Reviewer #2 (Public review): https://doi.org/10.7554/eLife.93695.3.sa2
Reviewer #3 (Public review): https://doi.org/10.7554/eLife.93695.3.sa3
Author response https://doi.org/10.7554/eLife.93695.3.sa4

## Additional files

### Supplementary files
• MDAR checklist

### Data availability

The benchmark datasets of protein binding sites are available in Zenodo, at https://doi.org/10.5281/zenodo.10845362. The source code and trained models of GPSite are available at https://github.com/biomed-AI/GPSite (copy archived at *Yuan, 2024*). The user-friendly GPSite webserver is freely available at https://bio-web1.nscc-gz.cn/app/GPSite. All predicted structures along with the binding site annotations for the Swiss-Prot sequences are available in our GPSiteDB database at https://bio-web1.nscc-gz.cn/database/GPSiteDB.

The following dataset was generated:

| Author(s) | Year | Dataset title | Dataset URL | Database and Identifier |
|---|---|---|---|---|
| Yuan Q, Yang Y | 2024 | Data from: Genome-scale annotation of protein binding sites via language model and geometric deep learning | https://doi.org/10.5281/zenodo.10845362 | Zenodo, 10.5281/zenodo.10845362 |

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

## Appendix 1

### Brief introductions to the competitive methods

#### DRNApred (*Yan and Kurgan, 2017*)

DRNApred is a sequence-based DNA- and RNA-binding site predictor, which is based on a logistic regression input with sequence features including evolutionary information, that is hidden Markov models (HMM) profile and predicted structural properties including secondary structure (SS), relative solvent accessibility (RSA), and disorder. We used its webserver (http://biomine.cs.vcu.edu/servers/DRNApred/) for performance evaluation.

#### NCBRPred (*Zhang et al., 2021*)

NCBRPred is a sequence-based DNA- and RNA-binding site predictor, which is based on a bidirectional Gated Recurrent Units (BiGRUs) input with sequence features including the evolutionary information PSSM (position-specific scoring matrix) and HMM, as well as the predicted RSA and SS. We used its webserver (http://bliulab.net/NCBRPred/server) for performance evaluation.

#### NucBind, SVMnuc, and COACH-D

NucBind (*Su et al., 2019*) is a structure-based DNA- and RNA-binding site predictor, which combines the predictions from a support vector machine (SVM) based ab-initio method SVMnuc and a template-based method COACH-D (*Wu et al., 2018*). SVMnuc was trained with sequence features from PSSM, HMM and predicted SS. We used its webserver (http://yanglab.nankai.edu.cn/NucBind/) for evaluation.

#### GraphBind (*Xia et al., 2021*)

GraphBind is a structure-based nucleic-acid- and ligand-binding site predictor, which adopts a hierarchical graph neural network (GNN) for massage passing on protein residue graphs. Its input features mainly contain PSSM, HMM, SS, and atomic features. We used its standalone program with pre-trained model weights from http://www.csbio.sjtu.edu.cn/bioinf/GraphBind/sourcecode.html for inference and evaluation.

#### GeoBind (*Li and Liu, 2023*)

GeoBind is a structure-based nucleic-acid- and ligand-binding site predictor, which employs geodesic convolution to point cloud on the protein surface. Its input features contain HMM, atom type, and local curvature of the surface. We used its webserver (http://www.zpliulab.cn/GeoBind/) for evaluation.

#### GraphSite (*Yuan et al., 2022b*)

GraphSite is a sequence-based DNA-binding site predictor, which adopts a graph transformer for massage passing on protein residue graphs constructed from AlphaFold2-predicted structures. Its input features contain sequence representations from AlphaFold2, PSSM, HMM, and structural properties (RSA, SS and torsion angles) from DSSP. We used its standalone program with pre-trained model weights from https://github.com/biomed-AI/GraphSite (*Yuan, 2023a*), for inference and evaluation.

#### aaRNA (*Li et al., 2014*)

aaRNA is a structure-based RNA-binding site predictor, which employs a fully connected neural network input with sequence features based on PSSM and HMM, as well as structural features like RSA, SS, and curvature of the protein surface. We used its webserver (https://sysimm.ifrec.osaka-u.ac.jp/aarna/) for evaluation.

#### PepBind (*Zhao et al., 2018*)

PepBind is a sequence-based peptide-binding site predictor, which combines the predictions from a SVM-based ab-initio method SVMpep and two template-based methods S-SITE and TM-SITE (*Yang et al., 2013*). SVMnuc was trained with sequence features from PSSM, HMM, predicted SS and predicted intrinsic disorder. The structures required by TM-SITE are predicted by the I-TASSER Suite (*Yang et al., 2015*). We used its webserver (http://yanglab.nankai.edu.cn/PepBind/) for evaluation.

## PepNN-Struct and PepNN-Seq (*Abdin et al., 2022*)

PepNN-Struct is a structure-based peptide-binding site predictor which employs a graph transformer network to encode the protein representations and applies reciprocal multi-head attention to model the interaction between the protein structure and peptide sequence. A one-hot encoding is used to represent the protein and peptide sequence information. The pre-trained contextualized language model ProtBert (*Elnaggar et al., 2022*) is also used to embed the protein sequences. To perform the peptide-agnostic binding site prediction, the model is input with random length poly-glycine peptides. PepNN-Seq is similar to PepNN-Struct, except that the graph transformer model is substituted with an MLP. We used its standalone program with pre-trained model weights from https://gitlab.com/oabdin/pepnn for inference and evaluation.

## PepBCL (*Wang et al., 2022*)

PepBCL is a sequence-based peptide-binding site predictor, which fine-tuned the pre-trained protein language model from *Elnaggar et al., 2022* to predict peptide-binding sites. It also used contrastive learning to address the data imbalance problem. We adopted its standalone code with pre-trained model weights for inference and evaluation (https://github.com/Ruheng-W/PepBCL, *Wang, 2023*). There are two PepBCL models trained on two different datasets (1154 vs 640 training proteins), and here we report the results of the model trained on the larger dataset, since it performs slightly better.

## DeepPPISP (*Zeng et al., 2020*)

DeepPPISP is a structure-based protein-protein binding site predictor, which utilizes one-hot protein sequence, PSSM, and SS as input. The model adopts a convolutional neural network (CNN) to capture local and global protein features. The predictions of DeepPPISP for the test proteins are directly obtained from our previous work (*Yuan et al., 2021*), which were originally produced by re-training DeepPPISP on our training set using its standalone code in https://github.com/CSUBioGroup/DeepPPISP (*CSUBioGroup, 2019*).

## SPPIDER (*Porollo and Meller, 2007*)

SPPIDER is a structure-based protein-protein binding site predictor, which is based on a fully connected neural network input with sequence features including PSSM and structure features like RSA. The model measures the impacts from spatially neighboring residues by adopting weighted averages over features of spatially nearest neighbors. The predictions of SPPIDER for the test proteins are directly obtained from our previous work (*Yuan et al., 2021*), which were originally generated by the SPPIDER webserver (https://sppider.cchmc.org/).

## MaSIF-site (*Gainza et al., 2020*)

MaSIF-site is a structure-based protein-protein binding site predictor, which maps the geometric and chemical features on the protein surface to patches and uses the geodesic convolutional layers to capture the surface fingerprints. MaSIF-site does not rely on any features from multiple sequence alignments (MSA). The predictions of MaSIF-site for the test proteins are directly obtained from our previous work (*Yuan et al., 2021*), which were originally generated by the standalone program with pre-trained model weights through a docker container from https://github.com/lpdi-epfl/masif (*Gainza, 2021*).

## GraphPPIS (*Yuan et al., 2021*)

GraphPPIS is a structure-based protein-protein binding site predictor, which exploits a deep graph convolutional neural network (GCN) with initial residual and identity mapping to refine information in the protein residue graphs. The input features of GraphPPIS consist of PSSM, HMM, and structural properties (RSA, SS and torsion angles) from DSSP. The prediction results for the test proteins can be obtained from our webserver (http://bio-web1.nscc-gz.cn/app/graphppis-v2) or the standalone program (https://github.com/biomed-AI/GraphPPIS, *Yuan, 2023b*).

## ScanNet (*Tubiana et al., 2022*)

ScanNet is a structure-based protein-protein binding site predictor, which adopts geometric deep learning for massage passing on protein atom graphs. Its input features mainly contain atomic features and PSSM. We used its standalone program with pre-trained model weights from https://github.com/jertubiana/ScanNet, *Tubiana, 2023* for inference and evaluation.

### PeSTo (*Krapp et al., 2023b*)

PeSTo is a structure-based protein-protein binding site predictor, which adopts a geometric transformer for massage passing on protein atom graphs. Its input feature only contains the atomic type. We used its standalone program with pre-trained model weights from https://github.com/LBM-EPFL/PeSTo (*Krapp, 2023a*) for inference and evaluation.

### TargetS (*Yu et al., 2013*)

TargetS is a sequence-based ligand-binding site predictor, which extracts evolutionary information (PSSM), predicted SS and ligand-specific binding propensity from sequence context using a sliding-window strategy. It then employs several SVMs to learn the local binding patterns, which are assembled by the modified AdaBoost algorithm. We used its webserver (http://www.csbio.sjtu.edu.cn/TargetS/) for evaluation.

### DELIA (*Xia et al., 2020*)

DELIA is a structure-based ligand-binding site predictor, which uses the bidirectional long short-term memory (BiLSTM) networks to refine sequence features including binding propensity from S-SITE, PSSM, HMM, predicted SS and predicted RSA, as well as a CNN to extract characteristics from the protein distance matrices. We used its webserver (http://www.csbio.sjtu.edu.cn/bioinf/delia/) for evaluation.

### MIB (*Lin et al., 2016*)

MIB is a template-based metal ion-binding site predictor, where the fragment transformation method is used for structural comparison between query proteins and templates without any data training. The predictions of MIB for the test proteins are directly obtained from our previous work (*Yuan et al., 2022c*), which were originally generated from the MIB webserver (http://bioinfo.cmu.edu.tw/MIB/).

### IonCom (*Hu et al., 2016*)

IonCom is a structure-based metal and acid radical ion-binding site predictor, which combines the predictions from an SVM-based ab-initio method IonSeq and four template-based methods including COFACTOR (*Roy et al., 2012*), TM-SITE, S-SITE, and COACH (*Yang et al., 2013*). IonSeq was trained with sequence features from PSSM, ligand-specific binding propensity, predicted SS, predicted RSA, etc. We used its standalone program with pre-trained weights from https://zhanggroup.org/IonCom/ for inference and evaluation.

### LMetalSite (*Yuan et al., 2022c*)

LMetalSite is a sequence-based alignment-free metal ion-binding site predictor where the pre-trained protein language model ProtTrans is used to extract sequence embeddings and a transformer with multi-task learning is applied to capture the intrinsic similarities between different metal ions. The prediction results of the test proteins can be obtained from our webserver (http://bio-web1.nscc-gz.cn/app/lmetalsite) or the standalone program (https://github.com/biomed-AI/LMetalSite, *Yuan, 2022a*).

## Performance comparison between GPSite and PeSTo

Since 340 out of 375 proteins in our protein-protein binding site test set share >30% identity with the training sequences of PeSTo, we performed a separate comparison between GPSite and PeSTo using the training and test datasets from PeSTo. By re-training with simply the same hyperparameters, GPSite achieves better performance than PeSTo (AUPR of 0.824 against 0.797) as shown in *Appendix 2—table 4*. Furthermore, when using ESMFold-predicted structures as input, the performance of PeSTo decreases substantially (AUPR of 0.691), and the superiority of our method will be further reflected. As in *Krapp et al., 2023b*, the performance of ScanNet is also included (AUPR of 0.720), which is also largely outperformed by GPSite.

## Case study for the ribosome biogenesis protein ERB1

Here we present an example of an RNA-binding protein, i.e., the ribosome biogenesis protein ERB1 (PDB: 7R6Q, chain m), to illustrate the impact of predicted structure's quality. As shown in *Appendix 3—figure 5*, ERB1 is an integral component of a large multimer structure comprising

protein and RNA chains (i.e. the state E2 nucleolar 60S ribosome biogenesis intermediate). Likely due to the neglect of interactions from other protein chains, ESMFold fails to predict the correct conformation of the ERB1 chain (TM-score=0.24). Using this incorrect predicted structure, GPSite achieves an AUPR of 0.580, lower than GraphBind input with the native structure (AUPR=0.636). However, the performance of GraphBind substantially declines to an AUPR of 0.468 when employing the predicted structure as input. Moreover, if GPSite adopts the native structure for prediction, a notable performance boost can be obtained (AUPR=0.681).

## Generation of the evolutionary features from MSA

Evolutionarily conserved residues may contain motifs related to important protein properties. Here, we also evaluated the widely used evolutionary features from MSA in our ablation studies, including position-specific scoring matrix (PSSM) and hidden Markov models (HMM) profile. PSSM is produced by running PSI-BLAST (*Altschul et al., 1997*) to search the query sequence against UniRef90 (*Suzek et al., 2007*) with three iterations and an E-value of 0.001. HMM profile is generated by running HHblits (*Remmert et al., 2011*) against UniClust30 (*Mirdita et al., 2017*) with default parameters. Each residue is encoded into a 20-dimensional vector in PSSM or HMM. The feature values in the sequence representations from PSSM and HMM are further normalized to scores between 0 and 1 as follows:

$$v_{norm} = \frac{v - v_{min}}{v_{max} - v_{min}} \tag{A1}$$

where $v$ is the original feature value, and $v_{min}$ and $v_{max}$ are the minimum and maximum values of this feature type observed in the training set, respectively.

## The effect of training with predicted structures

We examined the performance under different training and evaluation settings as shown in *Appendix 2—table 9*. As expected, the model yields exceptional performance (average AUPR of 0.656) when trained and evaluated using native structures. However, if this model is fed with predicted structures of the test proteins, the performance substantially declines to an average AUPR of 0.573. This trend aligns with the observations for other structure-based methods as illustrated in *Figure 2*. More importantly, in the practical scenario where only predicted structures are available for the target proteins, training the model with predicted structures (i.e. GPSite) results in superior performance than training the model with native structures (average AUPR of 0.594 against 0.573), probably owing to the consistency between the training and testing data. For completeness, the results in *Appendix 3—figure 2* are also included where GPSite is tested with native structures (average AUPR of 0.637).

## The cross-type performance of the multi-task network in GPSite

We conducted cross-type evaluations by applying different ligand-specific MLPs in GPSite for the test sets of different ligands. As shown in *Appendix 2—table 10*, for each ligand-binding site test set, the corresponding ligand-specific network consistently achieves the best performance. This indicates that the ligand-specific MLPs have specifically learned the binding patterns of particular molecules. We also noticed that the cross-type performance is reasonable for the ligands sharing similar properties. For instance, the DNA-specific MLP exhibits a reasonable AUPR when predicting RNA-binding sites, and vice versa. Similar trends are also observed between peptide and protein, as well as among metal ions as expected. Interestingly, the cross-type performance between ATP and HEM is also acceptable, potentially attributed to their comparable molecular weights (507.2 and 616.5, respectively).

## GPSite is effective for completing the function annotations in Swiss-Prot

As depicted in *Figure 5A*, GPSite assigns relatively high prediction scores to the proteins without 'protein binding' function in the Swiss-Prot annotations, leading to a modest AUC value of 0.608 (*Figure 5B*). This may be ascribed to the fact that protein-protein interactions are ubiquitous in living organisms while the Swiss-Prot function annotations are incomplete. To support this hypothesis, we present two proteins as case studies, both sharing <20% sequence identity with the protein-binding training set of GPSite. The first case is Aminodeoxychorismate synthase component 2 from *Escherichia coli* (UniProt ID: P00903). GPSite confidently predicted this protein as a protein-binding protein with a high prediction score of 0.936. Notably, this protein was not annotated with the

'protein binding' function (GO:0005515) or any of its GO child terms in the Swiss-Prot database at the time of manuscript preparation (https://rest.uniprot.org/unisave/P00903?format=txt&versions=171, release: 2023-05-03). However, in the latest release of Swiss-Prot (https://rest.uniprot.org/unisave/P00903?format=txt&versions=174, release: 2023-11-08) during manuscript revision, this protein is annotated with the 'protein heterodimerization activity' function (GO:0046982), which is a child term of 'protein binding'. In fact, the heterodimerization activity of this protein has been validated through experiments in the year of 1996 (PMID: 8679677), indicating the potential incompleteness of the Swiss-Prot annotations. The other case is Hydrogenase-2 operon protein HybE from *Escherichia coli* (UniProt ID: P0AAN1), which was also predicted as a protein-binding protein by GPSite (score=0.909). Similarly, this protein was not annotated with the 'protein binding' function in the Swiss-Prot database at the time of manuscript preparation (https://rest.uniprot.org/unisave/P0AAN1?format=txt&versions=108). However, in the latest release of Swiss-Prot (https://rest.uniprot.org/unisave/P0AAN1?format=txt&versions=111), this protein is annotated with the 'preprotein binding' function (GO:0070678), which is a child term of 'protein binding'. In fact, the preprotein binding function of this protein has been validated through experiments in the year of 2003 (PMID: 12914940). These cases demonstrate the effectiveness of GPSite for completing the missing function annotations in Swiss-Prot.

## Evaluation metrics

Following the previous studies, we use recall (Rec), precision (Pre), accuracy (Acc), F1-score (F1), Matthews correlation coefficient (MCC), area under the receiver operating characteristic curve (AUC), and area under the precision-recall curve (AUPR) to evaluate the prediction performance:

$$Rec = \frac{TP}{TP + FN} \tag{A2}$$

$$Pre = \frac{TP}{TP + FP} \tag{A3}$$

$$Acc = \frac{TP + TN}{TP + TN + FP + FN} \tag{A4}$$

$$F1 = 2 \times \frac{Pre \times Rec}{Pre + Rec} \tag{A5}$$

$$MCC = \frac{TP \times TN - FN \times FP}{\sqrt{(TP + FP) \times (TP + FN) \times (TN + FP) \times (TN + FN)}} \tag{A6}$$

where true positives (TP) and true negatives (TN) denote the numbers of correctly predicted binding and non-binding residues, and false positives (FP) and false negatives (FN) denote the numbers of incorrectly predicted binding and non-binding residues, respectively. AUC and AUPR are independent of thresholds, thus reflecting the overall performance of a model. The other metrics are calculated using a threshold to convert the predicted binding probabilities to binary predictions. We go through 101 thresholds from 0 to 1 with an interval of 0.01, and select the best threshold that maximizes MCC on the validation sets. We adopt AUPR for hyperparameter selections as it is more sensitive and informative than AUC in imbalanced two-class classification tasks (*Saito and Rehmsmeier, 2015*; *Davis and Goadrich, 2006*).

# Appendix 2

**Appendix 2—table 1.** Statistics of the 10 binding site benchmark datasets used in this study.

| | Training set | | | Test set | | |
|---|---|---|---|---|---|---|
| Molecule type | Sequences | Residues | % of binding residues | Sequences | Residues | % of binding residues |
| DNA | 661 | 185,796 | 8.06 | 146 | 57,914 | 5.75 |
| RNA | 689 | 205,648 | 10.55 | 346 | 105,230 | 9.78 |
| Peptide | 1251 | 348,370 | 5.39 | 235 | 74,788 | 4.50 |
| Protein | 335 | 66,366 | 15.63 | 375 | 78,475 | 14.57 |
| ATP | 347 | 130,655 | 3.91 | 79 | 39,459 | 3.12 |
| HEM | 176 | 47,063 | 8.55 | 48 | 15,618 | 6.21 |
| $Zn^{2+}$ | 1646 | 474,855 | 1.63 | 211 | 56,020 | 1.85 |
| $Ca^{2+}$ | 1554 | 504,146 | 1.67 | 183 | 66,854 | 1.55 |
| $Mg^{2+}$ | 1729 | 575,732 | 1.10 | 235 | 88,806 | 1.01 |
| $Mn^{2+}$ | 547 | 181,699 | 1.41 | 57 | 20,419 | 1.10 |

Note: We combined the two test sets (Test_60 and Test_315) from **Yuan et al., 2021** to establish our final protein-protein binding site test set.

**Appendix 2—table 2.** The performance of GPSite on the five-fold cross-validation and independent test sets.

| Molecule type | Five-fold cross-validation | | Test set | |
|---|---|---|---|---|
| | AUC | AUPR | AUC | AUPR |
| DNA | 0.933 | 0.620 | 0.921 | 0.516 |
| RNA | 0.910 | 0.615 | 0.899 | 0.573 |
| Peptide | 0.858 | 0.406 | 0.836 | 0.345 |
| Protein | 0.819 | 0.491 | 0.836 | 0.484 |
| ATP | 0.960 | 0.688 | 0.975 | 0.714 |
| HEM | 0.963 | 0.778 | 0.971 | 0.802 |
| $Zn^{2+}$ | 0.984 | 0.808 | 0.981 | 0.859 |
| $Ca^{2+}$ | 0.901 | 0.515 | 0.921 | 0.565 |
| $Mg^{2+}$ | 0.889 | 0.379 | 0.892 | 0.370 |
| $Mn^{2+}$ | 0.964 | 0.734 | 0.974 | 0.709 |
| Average | 0.918 | 0.603 | 0.921 | 0.594 |

**Appendix 2—table 3.** Performance comparison of GPSite with state-of-the-art methods on the 10 binding site test sets.

| Test set | Method | Rec | Pre | Acc | F1 | MCC | AUC | AUPR |
|---|---|---|---|---|---|---|---|---|
| DNA | DRNApred | 0.258 | 0.159 | 0.879 | 0.197 | 0.140 | 0.698 | 0.129 |
| | COACH-D | 0.247 | 0.315 | 0.926 | 0.277 | 0.241 | 0.674 | 0.197 |
| | NCBRPred | 0.225 | 0.316 | 0.927 | 0.263 | 0.230 | 0.763 | 0.229 |
| | SVMnuc | 0.319 | 0.319 | 0.922 | 0.319 | 0.277 | 0.806 | 0.259 |
| | NucBind | 0.333 | 0.329 | 0.923 | 0.331 | 0.290 | 0.806 | 0.264 |
| | GraphBind | 0.607 | 0.355 | 0.914 | 0.448 | 0.422 | 0.884 | 0.424 |
| | GeoBind* | 0.481 | 0.427 | 0.933 | 0.452 | 0.417 | 0.891 | 0.416 |
| | GeoBind | 0.520 | 0.442 | 0.935 | 0.478 | 0.445 | 0.896 | 0.443 |
| | GraphSite | 0.493 | 0.450 | 0.936 | 0.470 | 0.437 | <u>0.910</u> | <u>0.455</u> |
| | GPSite | 0.463 | 0.525 | 0.945 | 0.492 | 0.464 | **0.921** | **0.516** |
| RNA | COACH-D | 0.073 | 0.210 | 0.882 | 0.108 | 0.071 | 0.463 | 0.111 |
| | DRNApred | 0.092 | 0.236 | 0.882 | 0.133 | 0.093 | 0.530 | 0.142 |
| | NucBind | 0.185 | 0.344 | 0.886 | 0.241 | 0.195 | 0.649 | 0.226 |
| | SVMnuc | 0.227 | 0.371 | 0.887 | 0.282 | 0.232 | 0.742 | 0.275 |
| | NCBRPred | 0.234 | 0.471 | 0.899 | 0.312 | 0.284 | 0.660 | 0.302 |
| | aaRNA | 0.422 | 0.360 | 0.870 | 0.389 | 0.318 | 0.803 | 0.359 |
| | GeoBind | 0.562 | 0.455 | 0.891 | 0.503 | 0.446 | 0.804 | 0.459 |
| | GraphBind* | 0.576 | 0.342 | 0.850 | 0.429 | 0.365 | 0.828 | 0.433 |
| | GraphBind | 0.633 | 0.400 | 0.871 | 0.491 | 0.436 | <u>0.861</u> | <u>0.506</u> |
| | GPSite | 0.557 | 0.541 | 0.910 | 0.549 | 0.499 | **0.899** | **0.573** |
| Peptide | PepNN-Seq | 0.289 | 0.153 | 0.896 | 0.200 | 0.158 | 0.729 | 0.128 |
| | PepBind | 0.062 | 0.576 | 0.956 | 0.112 | 0.178 | 0.655 | 0.148 |
| | PepNN-Struct* | 0.351 | 0.180 | 0.899 | 0.238 | 0.202 | 0.765 | 0.163 |
| | PepNN-Struct | 0.337 | 0.210 | 0.913 | 0.259 | 0.222 | <u>0.783</u> | 0.187 |
| | PepBCL | 0.168 | 0.389 | 0.951 | 0.234 | 0.233 | 0.758 | <u>0.222</u> |
| | GPSite | 0.257 | 0.481 | 0.954 | 0.335 | 0.330 | **0.836** | **0.345** |
| Protein | DeepPPISP | 0.607 | 0.211 | 0.612 | 0.314 | 0.157 | 0.657 | 0.258 |
| | SPPIDER | 0.603 | 0.309 | 0.746 | 0.409 | 0.292 | 0.778 | 0.375 |
| | MaSIF-site | 0.584 | 0.330 | 0.767 | 0.421 | 0.308 | 0.777 | 0.384 |
| | GraphPPIS | 0.670 | 0.320 | 0.745 | 0.434 | 0.328 | 0.794 | 0.422 |
| | ScanNet* | 0.551 | 0.361 | 0.792 | 0.436 | 0.326 | 0.788 | 0.399 |
| | ScanNet | 0.568 | 0.442 | 0.832 | 0.497 | 0.403 | <u>0.832</u> | <u>0.476</u> |
| | GPSite | 0.490 | 0.473 | 0.846 | 0.481 | 0.391 | **0.836** | **0.484** |

*Appendix 2—table 3 Continued on next page*

*Appendix 2—table 3 Continued*

| Test set | Method | Rec | Pre | Acc | F1 | MCC | AUC | AUPR |
|---|---|---|---|---|---|---|---|---|
| ATP | TargetS | 0.451 | 0.549 | 0.971 | 0.495 | 0.483 | 0.855 | 0.447 |
| | GraphBind | 0.529 | 0.473 | 0.967 | 0.499 | 0.483 | 0.901 | 0.503 |
| | GeoBind | 0.614 | 0.479 | 0.967 | 0.538 | 0.526 | <u>0.927</u> | 0.534 |
| | DELIA* | 0.452 | 0.669 | 0.976 | 0.539 | 0.538 | 0.914 | 0.545 |
| | DELIA | 0.453 | 0.689 | 0.977 | 0.547 | 0.548 | 0.918 | <u>0.559</u> |
| | GPSite | 0.618 | 0.742 | 0.981 | 0.675 | 0.668 | **0.975** | **0.714** |
| HEM | TargetS | 0.504 | 0.756 | 0.959 | 0.605 | 0.598 | 0.892 | 0.581 |
| | GraphBind | 0.733 | 0.505 | 0.939 | 0.598 | 0.578 | 0.926 | 0.638 |
| | DELIA | 0.604 | 0.670 | 0.957 | 0.636 | 0.614 | 0.928 | 0.664 |
| | GeoBind* | 0.646 | 0.625 | 0.954 | 0.635 | 0.611 | 0.920 | 0.659 |
| | GeoBind | 0.707 | 0.710 | 0.964 | 0.709 | 0.689 | <u>0.932</u> | <u>0.724</u> |
| | GPSite | 0.715 | 0.762 | 0.968 | 0.738 | 0.722 | **0.971** | **0.802** |
| $Zn^{2+}$ | MIB | 0.744 | 0.219 | 0.946 | 0.339 | 0.385 | 0.935 | 0.394 |
| | TargetS | 0.454 | 0.749 | 0.987 | 0.566 | 0.578 | 0.874 | 0.593 |
| | IonCom* | 0.849 | 0.145 | 0.904 | 0.248 | 0.327 | 0.939 | 0.676 |
| | IonCom | 0.852 | 0.137 | 0.898 | 0.236 | 0.317 | 0.937 | 0.671 |
| | LMetalSite | 0.681 | 0.859 | 0.992 | 0.760 | 0.761 | <u>0.976</u> | <u>0.803</u> |
| | GPSite | 0.700 | 0.914 | 0.993 | 0.793 | 0.797 | **0.981** | **0.859** |
| $Ca^{2+}$ | MIB | 0.338 | 0.078 | 0.928 | 0.126 | 0.135 | 0.775 | 0.103 |
| | TargetS | 0.121 | 0.490 | 0.984 | 0.194 | 0.238 | 0.776 | 0.163 |
| | IonCom | 0.297 | 0.247 | 0.975 | 0.269 | 0.258 | 0.698 | 0.166 |
| | DELIA | 0.172 | 0.633 | 0.986 | 0.271 | 0.325 | 0.785 | 0.248 |
| | GeoBind | 0.279 | 0.515 | 0.985 | 0.362 | 0.372 | 0.895 | 0.348 |
| | GraphBind* | 0.290 | 0.537 | 0.985 | 0.377 | 0.388 | 0.836 | 0.335 |
| | GraphBind | 0.371 | 0.623 | 0.987 | 0.465 | 0.475 | 0.888 | 0.430 |
| | LMetalSite | 0.413 | 0.724 | 0.988 | 0.526 | 0.542 | <u>0.905</u> | <u>0.492</u> |
| | GPSite | 0.435 | 0.820 | 0.990 | 0.569 | 0.593 | **0.921** | **0.565** |
| $Mg^{2+}$ | MIB | 0.246 | 0.043 | 0.938 | 0.074 | 0.082 | 0.675 | 0.053 |
| | TargetS | 0.118 | 0.491 | 0.990 | 0.190 | 0.237 | 0.724 | 0.148 |
| | IonCom | 0.240 | 0.250 | 0.985 | 0.245 | 0.237 | 0.688 | 0.184 |
| | DELIA | 0.129 | 0.650 | 0.991 | 0.215 | 0.287 | 0.744 | 0.198 |
| | GeoBind | 0.181 | 0.475 | 0.990 | 0.263 | 0.289 | 0.840 | 0.227 |
| | GraphBind* | 0.246 | 0.205 | 0.983 | 0.224 | 0.216 | 0.750 | 0.136 |
| | GraphBind | 0.273 | 0.414 | 0.989 | 0.329 | 0.331 | 0.776 | 0.231 |
| | LMetalSite | 0.245 | 0.728 | 0.991 | 0.367 | 0.419 | <u>0.865</u> | <u>0.316</u> |
| | GPSite | 0.303 | 0.644 | 0.991 | 0.412 | 0.438 | **0.892** | **0.370** |

*Appendix 2—table 3 Continued on next page*

*Appendix 2—table 3 Continued*

| Test set | Method | Rec | Pre | Acc | F1 | MCC | AUC | AUPR |
|---|---|---|---|---|---|---|---|---|
| | MIB | 0.462 | 0.096 | 0.946 | 0.159 | 0.193 | 0.856 | 0.168 |
| | IonCom | 0.511 | 0.245 | 0.977 | 0.331 | 0.344 | 0.833 | 0.304 |
| | TargetS | 0.271 | 0.496 | 0.989 | 0.351 | 0.362 | 0.864 | 0.322 |
| | GeoBind | 0.569 | 0.479 | 0.988 | 0.520 | 0.516 | 0.938 | 0.454 |
| $Mn^{2+}$ | DELIA | 0.502 | 0.665 | 0.992 | 0.572 | 0.574 | 0.902 | 0.489 |
| | GraphBind* | 0.378 | 0.644 | 0.991 | 0.476 | 0.489 | 0.928 | 0.473 |
| | GraphBind | 0.427 | 0.706 | 0.992 | 0.532 | 0.545 | 0.930 | 0.555 |
| | LMetalSite | 0.613 | 0.719 | 0.993 | 0.662 | 0.661 | <u>0.966</u> | <u>0.625</u> |
| | GPSite | 0.613 | 0.807 | 0.994 | 0.697 | 0.701 | **0.974** | **0.709** |

Note: The best/second-best AUC and AUPR values are indicated by bold/underlined fonts. For the best experimental structure-based method (measured by AUPR) in each test set, its corresponding result when using ESMFold-predicted structures as input is denoted with *.

**Appendix 2—table 4.** Performance comparison of GPSite with ScanNet and PeSTo on the protein-protein binding site test set from PeSTo (*Krapp et al., 2023b*).

| Method | AUPR | AUC | MCC |
|---|---|---|---|
| ScanNet | 0.720 | 0.897 | 0.510 |
| PeSTo* | 0.691 | 0.886 | 0.451 |
| PeSTo | <u>0.797</u> | <u>0.929</u> | <u>0.636</u> |
| GPSite | **0.824** | **0.942** | **0.637** |

Note: The performance of ScanNet and PeSTo are directly obtained from *Krapp et al., 2023b*. PeSTo* denotes evaluation using the ESMFold-predicted structures as input. The metrics provided are the median AUPR, median AUC and median MCC. The best/second-best results are indicated by bold/underlined fonts.

**Appendix 2—table 5.** The numbers of proteins with TM-score >0.7 or ≤0.7 between native and ESMFold-predicted structures in the 10 binding site datasets.

| | Training set | | Test set | |
|---|---|---|---|---|
| Molecule type | >0.7 | ≤0.7 | >0.7 | ≤0.7 |
| DNA | 520 | 141 | 104 | 42 |
| RNA | 428 | 261 | 175 | 171 |
| Peptide | 1074 | 177 | 175 | 60 |
| Protein | 293 | 42 | 321 | 54 |
| ATP | 314 | 33 | 62 | 17 |
| HEM | 159 | 17 | 43 | 5 |
| $Zn^{2+}$ | 1428 | 218 | 160 | 51 |
| $Ca^{2+}$ | 1377 | 177 | 150 | 33 |
| $Mg^{2+}$ | 1565 | 164 | 195 | 40 |
| $Mn^{2+}$ | 512 | 35 | 52 | 5 |

**Appendix 2—table 6.** The prediction quality of ESMFold measured by TM-score between native and predicted structures in the 10 binding site datasets.

| Molecule type | Training set | | Test set | | Total | |
|---|---|---|---|---|---|---|
| | Median | Mean | Median | Mean | Median | Mean |
| DNA | 0.90 | 0.82 | 0.88 | 0.79 | 0.89 | 0.82 |
| RNA | 0.79 | 0.73 | 0.70 | 0.65 | 0.76 | 0.70 |
| Peptide | 0.93 | 0.86 | 0.88 | 0.78 | 0.93 | 0.85 |
| Protein | 0.94 | 0.87 | 0.93 | 0.85 | 0.93 | 0.86 |
| ATP | 0.95 | 0.89 | 0.90 | 0.83 | 0.94 | 0.88 |
| HEM | 0.95 | 0.89 | 0.94 | 0.87 | 0.94 | 0.88 |
| $Zn^{2+}$ | 0.94 | 0.87 | 0.91 | 0.82 | 0.93 | 0.86 |
| $Ca^{2+}$ | 0.95 | 0.88 | 0.93 | 0.85 | 0.94 | 0.88 |
| $Mg^{2+}$ | 0.95 | 0.90 | 0.93 | 0.86 | 0.95 | 0.89 |
| $Mn^{2+}$ | 0.96 | 0.92 | 0.95 | 0.91 | 0.96 | 0.91 |

**Appendix 2—table 7.** The ablation studies on protein features and model designs in the 10 binding site test sets.

| Method | DNA | RNA | Pep | Pro | ATP | HEM | $Zn^{2+}$ | $Ca^{2+}$ | $Mg^{2+}$ | $Mn^{2+}$ | Avg |
|---|---|---|---|---|---|---|---|---|---|---|---|
| w/o sequence | 0.389 | 0.473 | 0.251 | 0.396 | 0.646 | 0.726 | 0.791 | 0.503 | 0.338 | 0.646 | 0.516 |
| One-hot | 0.429 | 0.506 | 0.254 | 0.427 | 0.645 | 0.755 | 0.840 | 0.564 | 0.359 | 0.673 | 0.545 |
| MSA profile | 0.507 | 0.557 | 0.281 | 0.463 | 0.671 | 0.791 | 0.814 | 0.540 | 0.369 | 0.683 | 0.568 |
| w/o structure | 0.437 | 0.503 | 0.242 | 0.394 | 0.544 | 0.565 | 0.793 | 0.468 | 0.288 | 0.607 | 0.484 |
| w/o geometry | 0.484 | 0.539 | 0.318 | 0.439 | 0.631 | 0.670 | 0.813 | 0.489 | 0.313 | 0.638 | 0.533 |
| Single-task | 0.506 | 0.549 | 0.338 | 0.455 | 0.669 | 0.716 | 0.843 | 0.557 | 0.326 | 0.632 | 0.559 |
| GPSite | **0.516** | **0.573** | **0.345** | **0.484** | **0.714** | **0.802** | **0.859** | **0.565** | **0.370** | **0.709** | **0.594** |

Note: The numbers in this table are AUPR values. Bold fonts indicate the best results. 'Pep' and 'Pro' denote peptide and protein, respectively. 'Avg' means the average AUPR values among the 10 test sets. 'One-hot' denotes replacing the ProtTrans embedding with one-hot sequence encoding. The generation of the MSA profile (PSSM and HMM) is detailed in Generation of the evolutionary features from MSA. 'w/o structure' means using a transformer model only input with the ProtTrans sequence features. 'w/o geometry' means removing the geometric featurizer in GPSite.

**Appendix 2—table 8.** Performance comparison between GPSite and the baseline model using MSA profile for proteins with different Neff values in the combined test set of the 10 ligands.

| Neff | Sequences | Residues | MSA AUC | GPSite AUC | p-value |
|---|---|---|---|---|---|
| [1, 2) | 67 | 18,236 | 0.818 | 0.850 | $4.3 \times 10^{-8}$ |
| [2, 3) | 32 | 9395 | 0.856 | 0.854 | 0.72 |
| [3, 4) | 71 | 18,328 | 0.895 | 0.894 | 0.13 |
| [4, 5) | 133 | 30,392 | 0.901 | 0.896 | $4.0 \times 10^{-4}$ |
| [5, 6) | 182 | 39,858 | 0.909 | 0.916 | $9.8 \times 10^{-4}$ |
| [6, 7) | 226 | 60,128 | 0.915 | 0.913 | 0.10 |
| [7, 8) | 257 | 92,791 | 0.920 | 0.931 | $1.1 \times 10^{-9}$ |
| [8, +∞) | 947 | 334,455 | 0.919 | 0.935 | $7.0 \times 10^{-10}$ |

Note: Significance tests are performed following the procedure in *Yan and Kurgan, 2017*; *Xia et al., 2021*. If p-value <0.05, the difference between the performance is considered statistically significant.

**Appendix 2—table 9.** Performance comparison on the 10 binding site test sets under different training and evaluation settings.

| Setting | DNA | RNA | Pep | Pro | ATP | HEM | $Zn^{2+}$ | $Ca^{2+}$ | $Mg^{2+}$ | $Mn^{2+}$ | Avg |
|---|---|---|---|---|---|---|---|---|---|---|---|
| Train: native Test: native | 0.587 | 0.634 | 0.368 | 0.552 | 0.746 | 0.846 | 0.905 | 0.705 | 0.428 | 0.786 | 0.656 |
| Train: native Test: predicted | 0.497 | 0.554 | 0.311 | 0.459 | 0.704 | 0.784 | 0.826 | 0.546 | 0.352 | 0.694 | 0.573 |
| Train: predicted Test: native | 0.554 | 0.610 | 0.371 | 0.529 | 0.733 | 0.844 | 0.890 | 0.660 | 0.415 | 0.761 | 0.637 |
| Train: predicted Test: predicted (GPSite) | 0.516 | 0.573 | 0.345 | 0.484 | 0.714 | 0.802 | 0.859 | 0.565 | 0.370 | 0.709 | 0.594 |

Note: The numbers in this table are AUPR values. 'Pep' and 'Pro' denote peptide and protein, respectively. 'Avg' means the average AUPR values among the 10 test sets. 'native' and 'predicted' denote applying native and predicted structures as input, respectively.

**Appendix 2—table 10.** Cross-type performance by applying different ligand-specific MLPs in GPSite for the test sets of different ligands.

| Ligand-specific MLP | Ligand-binding site test set | | | | | | | | | |
|---|---|---|---|---|---|---|---|---|---|---|
| | DNA | RNA | Pep | Pro | ATP | HEM | $Zn^{2+}$ | $Ca^{2+}$ | $Mg^{2+}$ | $Mn^{2+}$ |
| DNA | **0.516** | <u>0.461</u> | 0.158 | 0.327 | 0.123 | 0.425 | 0.032 | 0.033 | 0.028 | 0.072 |
| RNA | <u>0.381</u> | **0.573** | 0.170 | 0.332 | 0.189 | 0.549 | 0.038 | 0.049 | 0.037 | 0.093 |
| Pep | 0.170 | 0.199 | **0.345** | <u>0.410</u> | 0.089 | 0.479 | 0.046 | 0.027 | 0.028 | 0.080 |
| Pro | 0.187 | 0.214 | 0.201 | **0.484** | 0.031 | 0.117 | 0.030 | 0.026 | 0.015 | 0.025 |
| ATP | 0.193 | 0.319 | 0.165 | 0.296 | **0.714** | <u>0.762</u> | 0.036 | 0.076 | 0.062 | 0.138 |
| HEM | 0.231 | 0.316 | <u>0.236</u> | 0.321 | <u>0.544</u> | **0.802** | 0.073 | 0.026 | 0.040 | 0.086 |
| $Zn^{2+}$ | 0.076 | 0.164 | 0.069 | 0.197 | 0.077 | 0.115 | **0.859** | 0.136 | 0.111 | <u>0.622</u> |
| $Ca^{2+}$ | 0.091 | 0.197 | 0.079 | 0.234 | 0.151 | 0.074 | 0.114 | **0.565** | 0.317 | 0.460 |
| $Mg^{2+}$ | 0.117 | 0.206 | 0.091 | 0.232 | 0.265 | 0.208 | 0.192 | <u>0.468</u> | **0.370** | 0.597 |
| $Mn^{2+}$ | 0.108 | 0.196 | 0.095 | 0.226 | 0.245 | 0.237 | <u>0.627</u> | 0.390 | <u>0.321</u> | **0.709** |

Note: 'Pep' and 'Pro' denote peptide and protein, respectively. The numbers in this table are AUPR values. The best/second-best result in each test set is indicated by bold/underlined font.

## Appendix 3

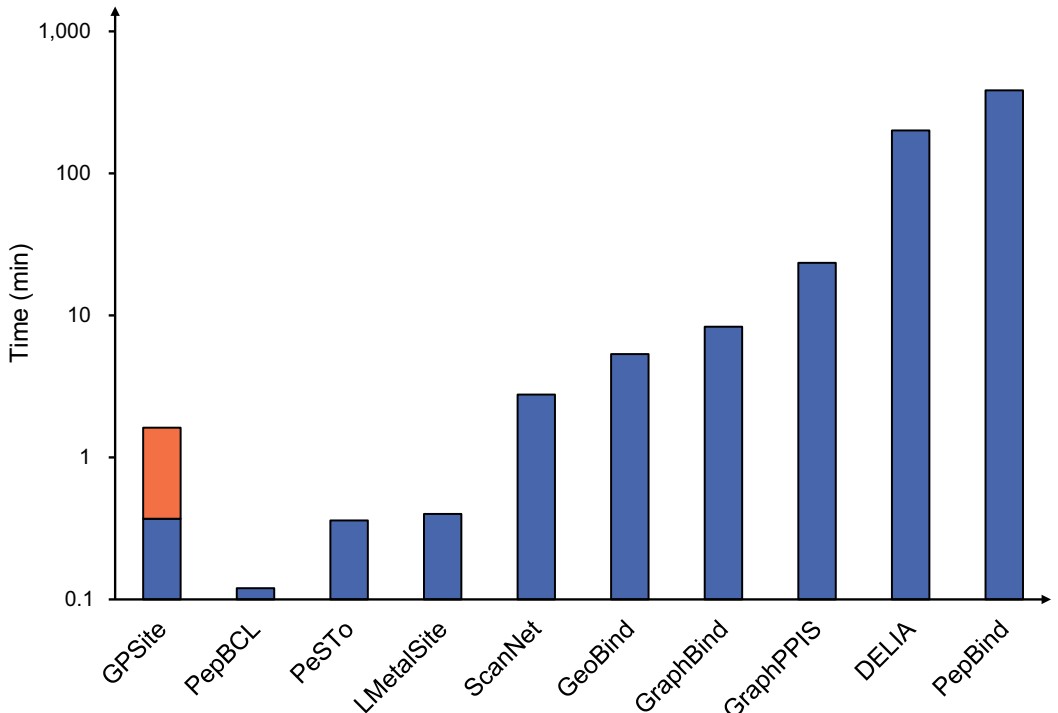

**Appendix 3—figure 1.** Runtime comparison of the GPSite webserver with other top-performing servers. Five protein chains (i.e. 8HN4_B, 8USJ_A, 8C1U_A, 8K3V_A, and 8EXO_A) comprising 100, 300, 500, 700, and 900 residues, respectively, were selected for testing, and the average runtime is reported for each method. Note that a significant portion of GPSite's runtime (75 s, indicated in orange) is allocated to structure prediction using ESMFold.

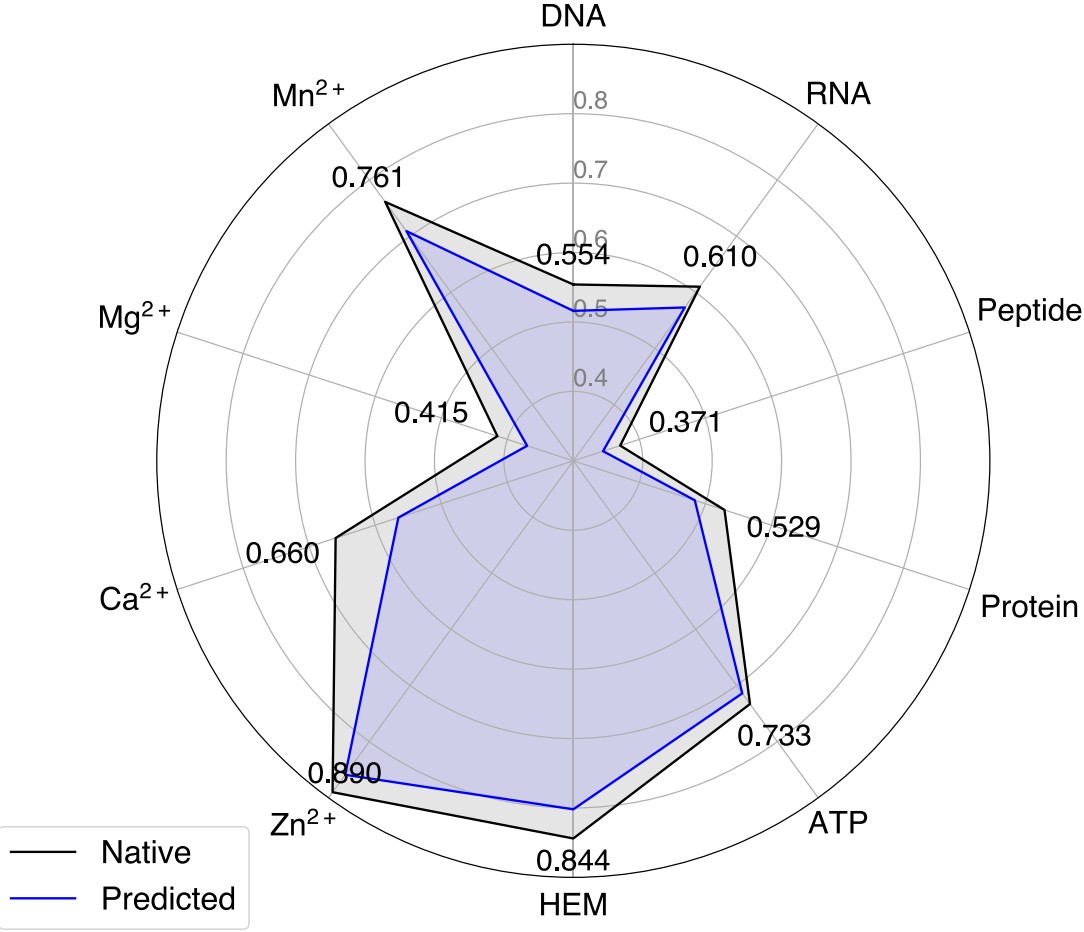

**Appendix 3—figure 2.** The performance of GPSite when using native or predicted structures as input during the test phase.

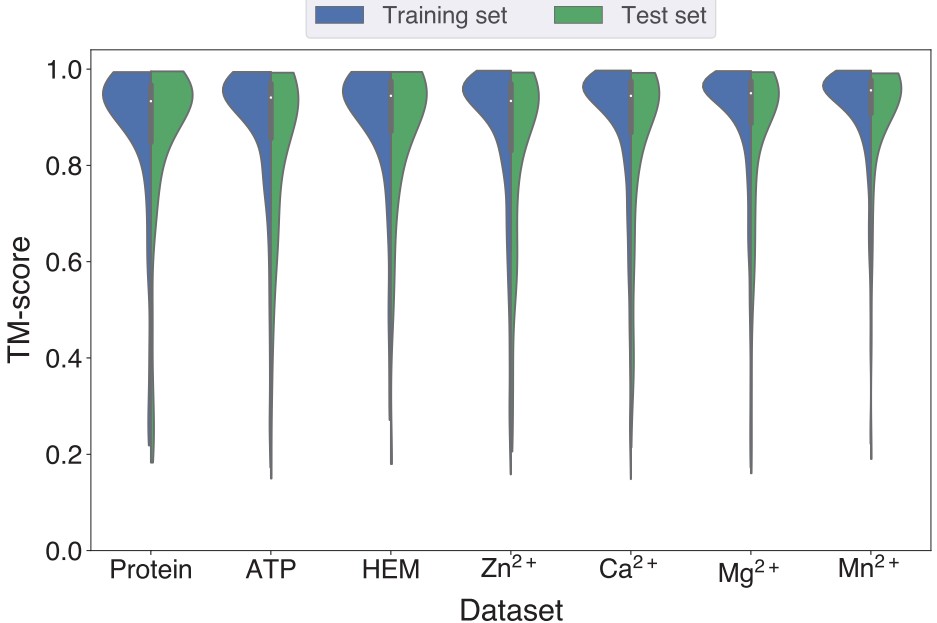

**Appendix 3—figure 3.** Distributions of the TM-scores between native and predicted structures in the protein, ATP, HEM, $Zn^{2+}$, $Ca^{2+}$, $Mg^{2+}$, and $Mn^{2+}$ datasets.

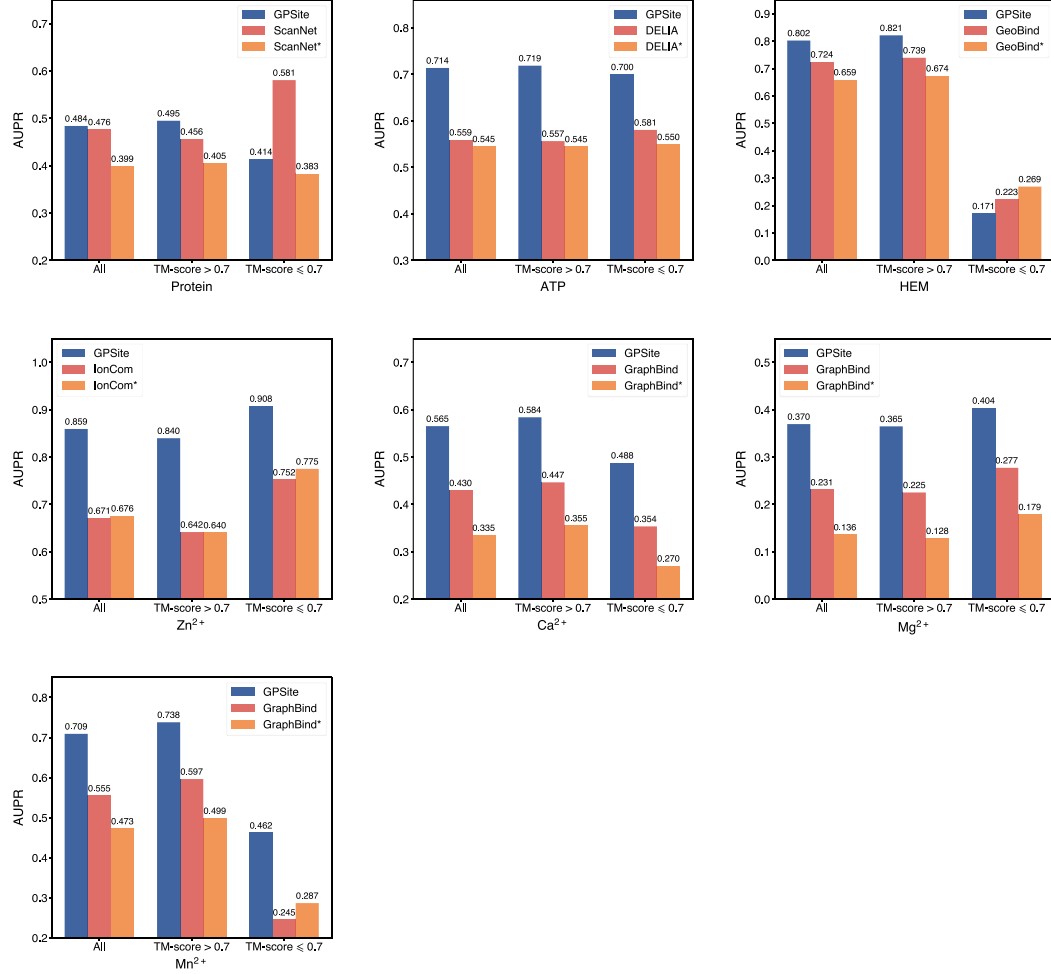

**Appendix 3—figure 4.** The performance of GPSite on structures of different qualities, and the comparisons with the best experimental structure-based methods in the test sets of protein, ATP, HEM, $Zn^{2+}$, $Ca^{2+}$, $Mg^{2+}$, and $Mn^{2+}$. The experimental structure-based methods input with ESMFold-predicted structures are marked with *. Since there are only 5 proteins with TM-score ≤0.7 in the HEM and $Mn^{2+}$ test sets (details shown in **Appendix 2—table 5**), the corresponding results may not be statistically significant.

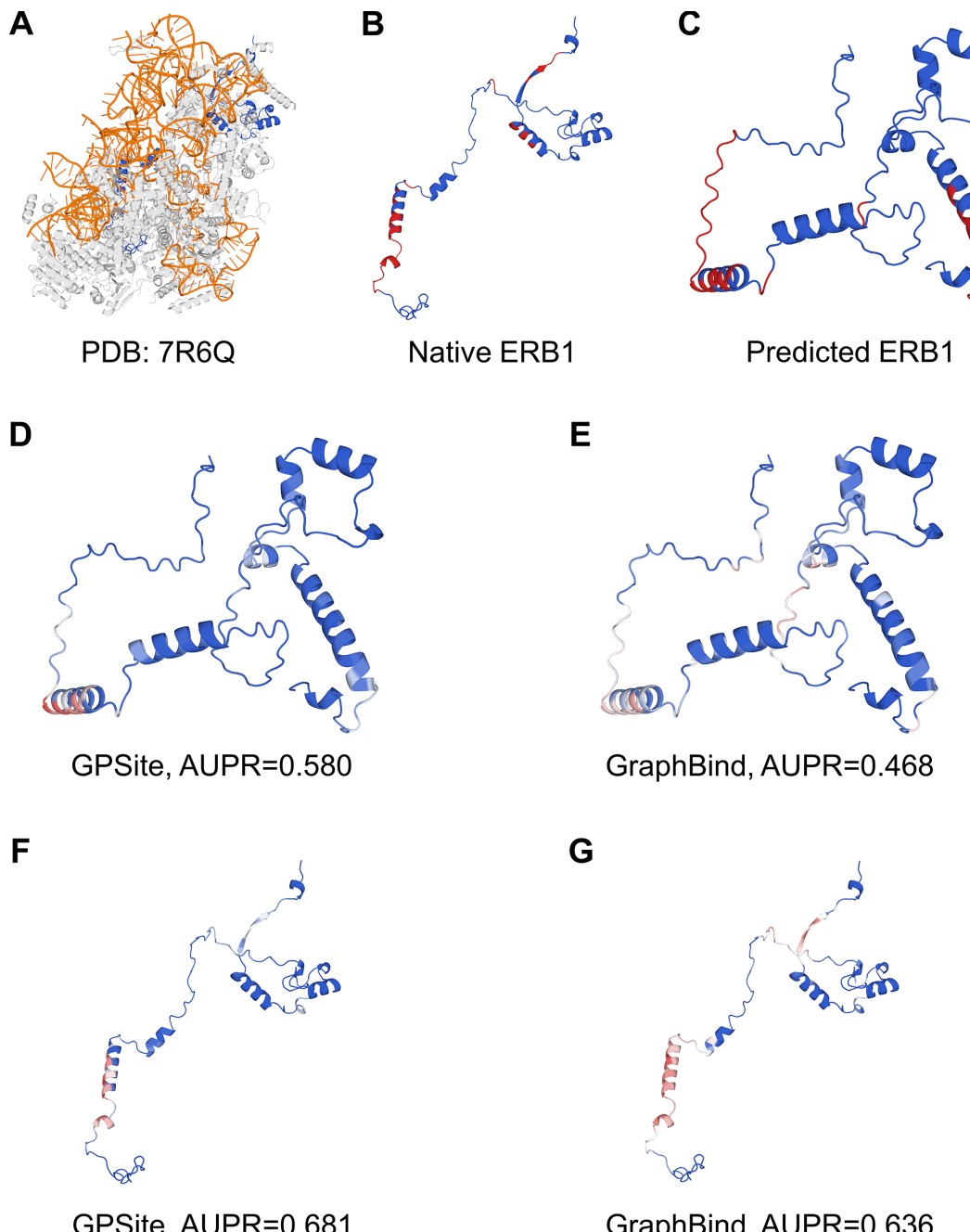

**Appendix 3—figure 5.** The prediction results of GPSite and GraphBind for the ribosome biogenesis protein ERB1. (**A**) The state E2 nucleolar 60S ribosome biogenesis intermediate (PDB: 7R6Q). The ribosome biogenesis protein ERB1 (chain m) is highlighted in blue, while other protein chains are colored in gray. The RNA chains are shown in orange. (**B**) The RNA-binding sites on ERB1 (colored in red). (**C**) The ESMFold-predicted structure of ERB1 (TM-score=0.24). The RNA-binding sites are also mapped onto this predicted structure (colored in red). (**D–G**) The prediction results of GPSite and GraphBind for the predicted and native ERB1 structures. The confidence of the predictions is represented with a gradient of color from blue for non-binding to red for binding.

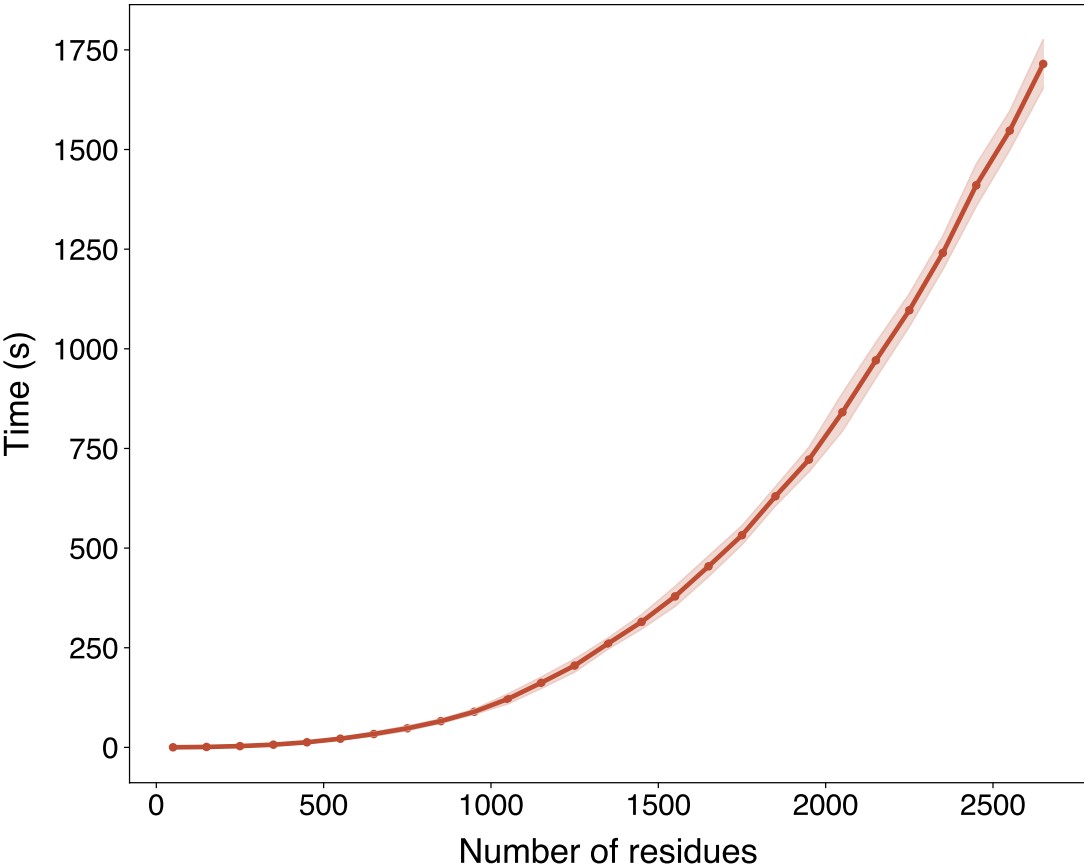

**Appendix 3—figure 6.** The run time of ESMFold with respect to the sequence length in Swiss-Prot evaluated on an NVIDIA A100 GPU. The run time is presented as mean ± standard deviation per range of number of residues (range size equals 100).

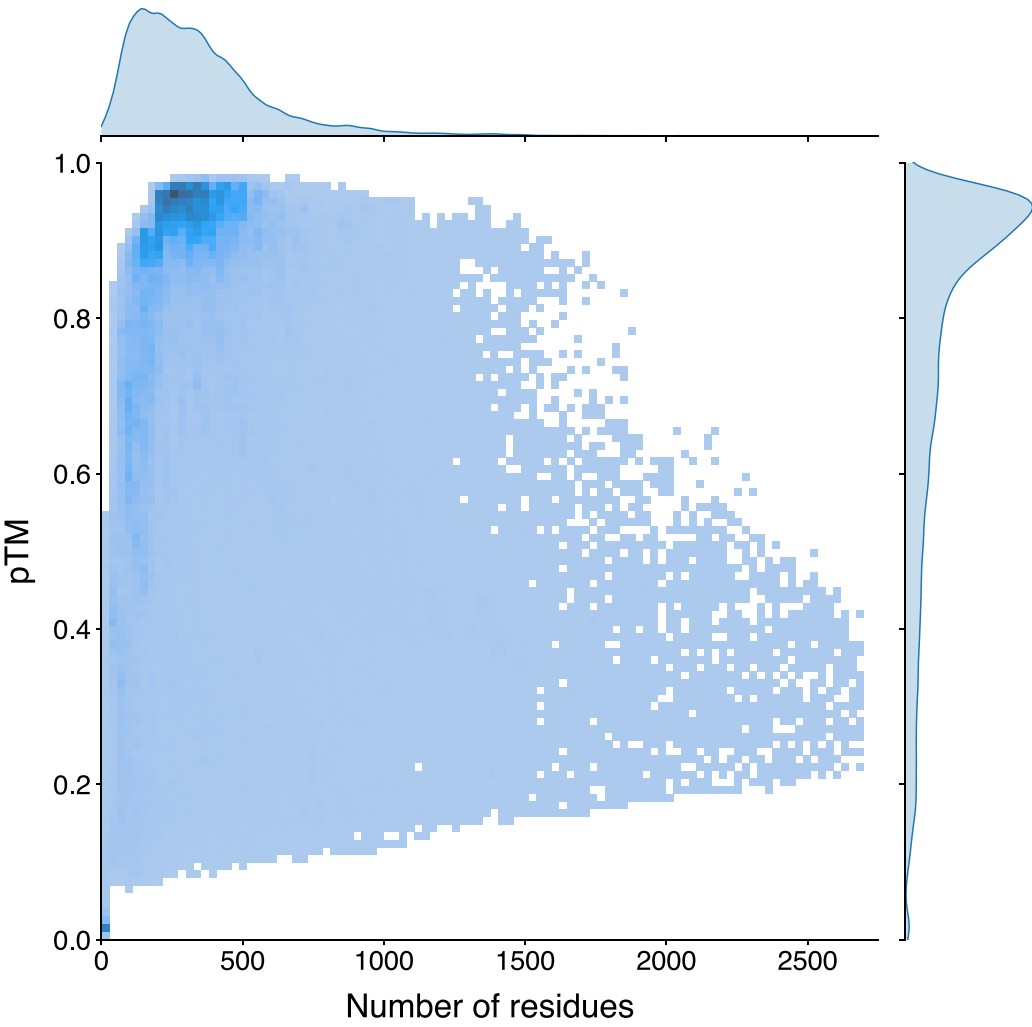

**Appendix 3—figure 7.** The univariate and bivariate distributions of the protein length and the pTM estimated by ESMFold of the Swiss-Prot sequences. The probability density curves are fit using kernel density estimation. The darker region in the bivariate heatmap corresponds to a higher number of samples.

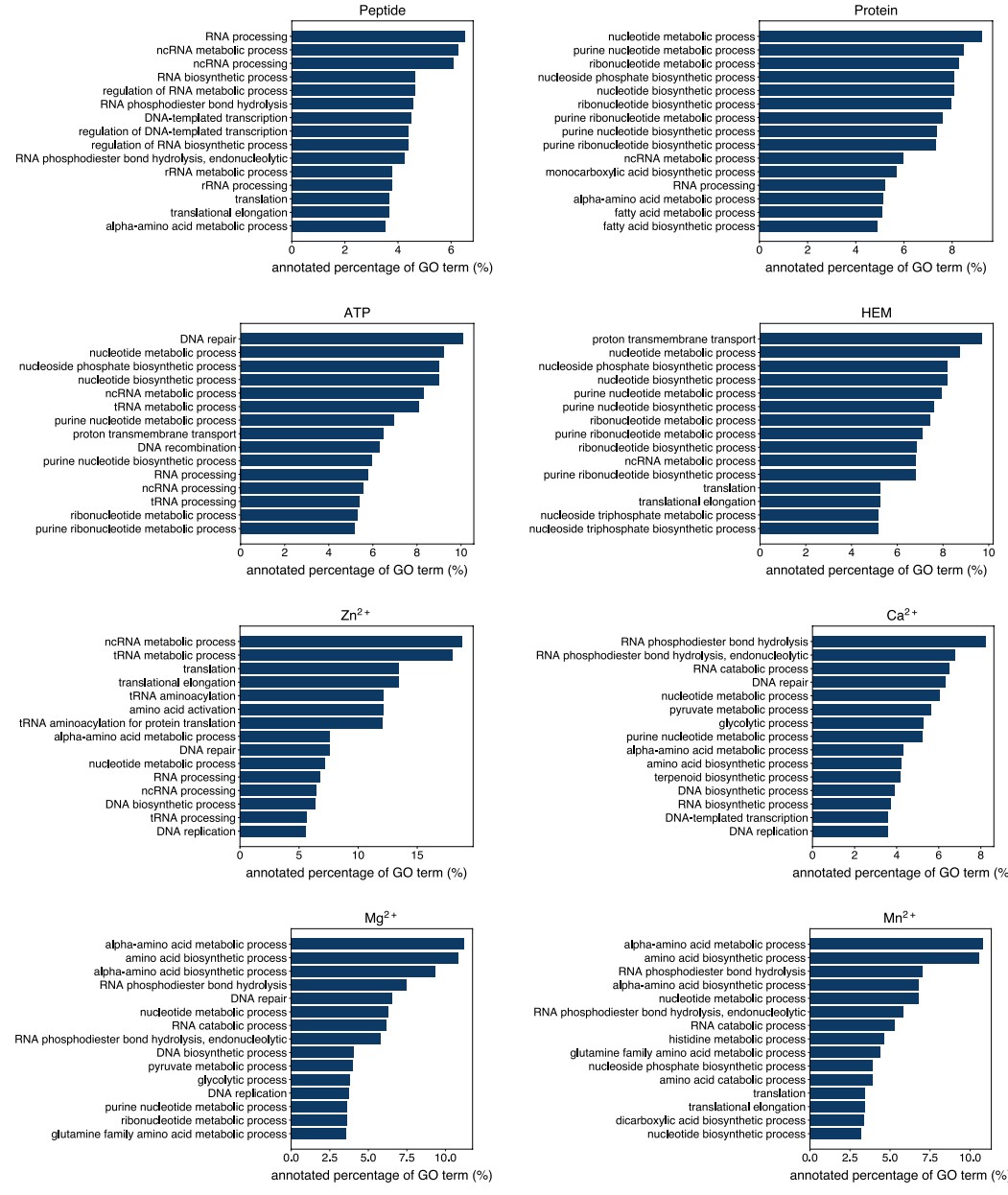

**Appendix 3—figure 8.** The percentage of proteins predicted as binding to peptide, protein, ATP, HEM, $Zn^{2+}$, $Ca^{2+}$, $Mg^{2+}$ and $Mn^{2+}$ by GPSite to be annotated with certain biological process in Swiss-Prot. Only the specific biological process terms with depth ≥8 in the GO directed acyclic graph are considered, among which the 15 terms with the highest percentage are displayed.

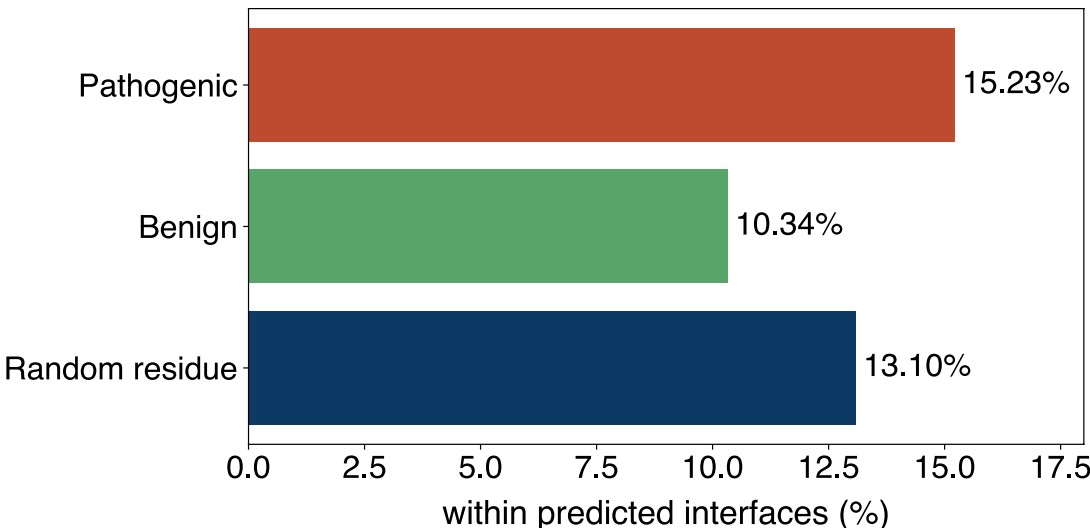

**Appendix 3—figure 9.** The percentage of pathogenic or benign natural variant sites within GPSite-predicted interfaces. The baseline is the probability of a random residue being annotated as an interface residue.

