## [Editor Report · eLife assessment]

The authors introduce a **valuable** machine-learning model for predicting binding sites of diverse ligands, including DNA, RNA, peptides, proteins, ATP, HEM, and metal ions, on proteins. The method is freely accessible and user-friendly. The authors have conducted thorough benchmarking and ablation studies, providing **convincing** evidence of the model's overall performance, despite some imperfections of the comparisons to other methods that arise from intrinsic differences between training methods and data.

---

## [Referee Report · Reviewer #1 (Public review)]

Summary:

The authors aim to address a critical challenge in the field of bioinformatics: the accurate and efficient identification of protein binding sites from sequences. Their work seeks to overcome the limitations of current methods, which largely depend on multiple sequence alignments or experimental protein structures, by introducing GPSite, a multi-task network designed to predict binding residues of various molecules on proteins using ESMFold.

Strengths:

(1) Benchmarking. The authors provide a comprehensive benchmark against multiple methods, showcasing the performances of a large number of methods in various scenarios.

(2) Accessibility and Ease of Use. GPSite is highlighted as a freely accessible tool with user-friendly features on their website, enhancing its potential for widespread adoption in the research community.

Weaknesses:

(1) Lack of significant insights. The paper reproduces results and analyses already presented in previous literature, without providing significant novel analysis or interpretation. However, they show a novel method with an original approach.

The work is useful for the field, especially in disease mechanism elucidation and novel drug design. The availability of genome-scale binding residue annotations GPSite offers is a significant advancement.

---

## [Referee Report · Reviewer #2 (Public review)]

Summary:

This work provides a new framework, "GPsite" to predict DNA, RNA, peptide, protein, ATP, HEM, and metal ions binding sites on proteins. This framework comes with a webserver and a database of annotations. The core of the model is a Geometric featurizer neural network that predicts the binding sites of a protein. One major contribution of the authors is the fact that they feed this neural network with predicted structure from ESMFold for training and prediction (instead of native structure in similar works) and a high-quality protein Language Model representation. The other major contribution is that it provides the public with a new light framework to predict protein-ligand interactions for a broad range of ligands. It is a convincing outcome of previous efforts to Geometric Deep Learning approaches to model protein-ligand interactions. The authors have demonstrated the interest of their framework with comprehensive ablation studies and benchmarks.

Strengths:

- The performance of this framework as well as the provided dataset and web server make it useful to conduct studies.

- The ablations of some core elements of the method, such as the protein Language Model part, the use of multiple ligands in the same model, the input structure, or the use of predicted structure to complement native structure are very insightful. They can help convince the reader that every part of the framework is necessary. This could also guide further developments in the field. As such, the presentation of this part of the work holds a critical place in this work.

Weaknesses:

- The authors made an important effort to compare their work to other similar frameworks. Yet, the lack of homogeneity of training methods and data from one work to the other makes the comparison slightly unconvincing, as the authors pointed out. Ablations performed by the authors were able to compensate for this general weakness, as well as the focus on several example structures.

---

## [Referee Report · Reviewer #3 (Public review)]

Summary

The authors of this work aim to address the challenge of accurately and efficiently identifying protein binding sites from sequences. They recognize that the limitations of current methods, including reliance on multiple sequence alignments or experimental protein structure, and the under-explored geometry of the structure, which limit the performance and genome-scale applications. The authors have developed a multi-task network, GPSite, that predicts binding residues for a range of biologically relevant molecules, including DNA, RNA, peptides, proteins, ATP, HEM, and metal ions, using sequence embeddings from protein language models and ESMFold-predicted structures. The reported results showed to be superior to current sequence-based and structure-based methods in terms of accuracy and efficiency.

Strengths

(1) The GPSite model's ability to predict binding sites for a wide variety of molecules, including DNA, RNA, peptides, and various metal ions.

(2) Based on the presented results, GPSite outperforms state-of-the-art methods in several benchmark datasets in terms of accuracy and efficiency.

(3) GPSite adopts predicted structure instead of native structures as input, enabling the model to be applied to a wider range of scenarios where native structures are rare.

(4) The low computational cost of GPSite is beneficial, which enables rapid genome-scale binding residue annotations, indicating the model's potential for large-scale downstream applications and discoveries.

Weaknesses

There are no major weaknesses after the revision.

---

## [Author Response]

The following is the authors’ response to the original reviews.

**Reviewer #1 (Public Review):**
Summary:The authors aim to address a critical challenge in the field of bioinformatics: the accurate and efficient identification of protein binding sites from sequences. Their work seeks to overcome the limitations of current methods, which largely depend on multiple sequence alignments or experimental protein structures, by introducing GPSite, a multi-task network designed to predict binding residues of various molecules on proteins using ESMFold.Strengths:Benchmarking. The authors provide a comprehensive benchmark against multiple methods, showcasing the performances of a large number of methods in various scenarios.Accessibility and Ease of Use. GPSite is highlighted as a freely accessible tool with user-friendly features on its website, enhancing its potential for widespread adoption in the research community.

RE: We thank the reviewer for acknowledging the contributions and strengths of our work!

Weaknesses:Lack of Novelty. The method primarily combines existing approaches and lacks significant technical innovation. This raises concerns about the original contribution of the work in terms of methodological development. Moreover, the paper reproduces results and analyses already presented in previous literature, without providing novel analysis or interpretation. This further diminishes the contribution of this paper to advancing knowledge in the field.

RE: The novelty of this work is primarily manifested in four key aspects. Firstly, although we have employed several existing tools such as ProtTrans and ESMFold to extract sequence features and predict protein conformations, these techniques were hardly explored in the field of binding site prediction. We have successfully demonstrated the feasibility of substituting multiple sequence alignments with language model embeddings and training with predicted structures, providing a new solution to overcome the limitations of current methods for genome-wide applications. Secondly, though a few methods tend to capture geometric information based on protein surfaces or atom graphs, surface calculation and property mapping are usually time-consuming, while massage passing on full atom graphs is memory-consuming and thus challenging to process long sequences. Besides, these methods are sensitive towards details and errors in the predicted structures. To facilitate large-scale annotations, we have innovatively applied geometric deep learning to protein residue graphs for comprehensively capturing backbone and sidechain geometric contexts in an efficient and effective manner (Figure 1). Thirdly, we have not only exploited multi-task learning to integrate diverse ligands and enhance performance, but also shown its capability to easily extend to the binding site prediction of other unseen ligands (Figure 4 D-E). Last but not least, as a “Tools and Resources” article, we have provided a fast, accurate and user-friendly webserver, as well as constructed a large annotation database for the sequences in Swiss-Prot. Leveraging this database, we have conducted extensive analyses on the associations between binding sites and molecular functions, biological processes, and disease-causing mutations (Figure 5), indicating the potential of our tool to unveil unexplored biology underlying genomic data.

We have now revised the descriptions in the “The geometry-aware protein binding site predictor (GPSite)” section to highlight the novelty of our work in a clearer manner:

“In conclusion, GPSite is distinguished from the previous approaches in four key aspects. First, profiting from the effectiveness and low computational cost of ProtTrans and ESMFold, GPSite is liberated from the reliance on MSA and native structures, thus enabling genome-wide binding site prediction. Second, unlike methods that only explore the C_α_ models of proteins ^25,40^, GPSite exploits a comprehensive geometric featurizer to fully refine knowledge in the backbone and sidechain atoms. Third, the employed message propagation on residue graphs is global structure-aware and time-efficient compared to the methods based on surface point clouds ^21,22^, and memory-efficient unlike methods based on full atom graphs ^23,24^. Residue-based message passing is also less sensitive towards errors in the predicted structures. Last but not least, instead of predicting binding sites for a single molecule type or learning binding patterns separately for different molecules, GPSite applies multi-task learning to better model the latent relationships among different binding partners.”

Benchmark Discrepancies. The variation in benchmark results, especially between initial comparisons and those with PeSTo. GPSite achieves a PR AUC of 0.484 on the global benchmark but a PR AUC of 0.61 on the benchmark against PeSTo. For consistency, PeSTo should be included in the benchmark against all other methods. It suggests potential issues with the benchmark set or the stability of the method. This inconsistency needs to be addressed to validate the reliability of the results.

RE: We thank the reviewer for the constructive comments. Since our performance comparison experiments involved numerous competitive methods whose training sets are disparate, it was difficult to compare or rank all these methods fairly using a single test set. Given the substantial overlap between our protein-binding site test set and the training set of PeSTo, we meticulously re-split our entire protein-protein binding site dataset to generate a new test set that avoids any overlap with the training sets of both GPSite and PeSTo and performed a separate evaluation, where GPSite achieves a higher AUPR than PeSTo (0.610 against 0.433). This is quite common in this field. For instance, in the study of PeSTo (Nat Commun 2023), the comparisons of PeSTo with MaSIF-site, SPPIDER, and PSIVER were conducted using one test set, while the comparison with ScanNet was performed on a separate test set.

Based on the reviewer’s suggestion, we have now replaced this experiment with a direct comparison with PeSTo using the datasets from PeSTo, in order to enhance the completeness and convincingness of our results. The corresponding descriptions are now added in Appendix 1-note 2, and the results are added in Appendix 2-table 4. For convenience, we also attach the note and table here:

“Since 340 out of 375 proteins in our protein-protein binding site test set share > 30% identity with the training sequences of PeSTo, we performed a separate comparison between GPSite and PeSTo using the training and test datasets from PeSTo. By re-training with simply the same hyperparameters, GPSite achieves better performance than PeSTo (AUPR of 0.824 against 0.797) as shown in Appendix 2-table 4. Furthermore, when using ESMFold-predicted structures as input, the performance of PeSTo decreases substantially (AUPR of 0.691), and the superiority of our method will be further reflected. As in ^24^, the performance of ScanNet is also included (AUPR of 0.720), which is also largely outperformed by GPSite.”

**Author response table 1. sa4table1:** Performance comparison of GPSite with ScanNet and PeSTo on the protein-protein binding site test set from PeSTo ^24^.

Method	AUPR	AUC	MCC
ScanNet	0.720	0.897	0.510
PeSTo^*^	0.691	0.886	0.451
PeSTo	0.797	0.929	0.636
GPSite	**0.824**	**0.942**	**0.637**

Note: The performance of ScanNet and PeSTo are directly obtained from ^24^. PeSTo^*^ denotes evaluation using the ESMFold-predicted structures as input. The metrics provided are the median AUPR, median AUC and median MCC. The best/second-best results are indicated by bold/underlined fonts.

Interface Definition Ambiguity. There is a lack of clarity in defining the interface for the binding site predictions. Different methods are trained using varying criteria (surfaces in MaSIF-site, distance thresholds in ScanNet). The authors do not adequately address how GPSite's definition aligns with or differs from these standards and how this issue was addressed. It could indicate that the comparison of those methods is unreliable and unfair.

RE: We thank the reviewer for the comments. The precise definition of ligand-binding sites is elucidated in the “Benchmark datasets” section. Specifically, the datasets of DNA, RNA, peptide, ATP, HEM and metal ions used to train GPSite were collected from the widely acknowledged BioLiP database [PMID: 23087378]. In BioLiP, a binding residue is defined if the smallest atomic distance between the target residue and the ligand is <0.5 Å plus the sum of the Van der Waal’s radius of the two nearest atoms. Meanwhile, most comparative methods regarding these ligands were also trained on data from BioLiP, thereby ensuring fair comparisons.

However, since BioLiP does not include data on protein-protein binding sites, studies for protein-protein binding site prediction may adopt slightly distinct label definitions, as the reviewer suggested. Here, we employed the protein-protein binding site data from our previous study [PMID: 34498061], where a protein-binding residue was defined as a surface residue (relative solvent accessibility > 5%) that lost more than 1 Å^2^ absolute solvent accessibility after protein-protein complex formation. This definition was initially introduced in PSIVER [PMID: 20529890] and widely applied in various studies (e.g., PMID: 31593229, PMID: 32840562). SPPIDER [PMID: 17152079] and MaSIF-site [PMID: 31819266] have also adopted similar surface-based definitions as PSIVER. On the other hand, ScanNet [PMID: 35637310] employed an atom distance threshold of 4 Å to define contacts while PeSTo [PMID: 37072397] used a threshold of 5 Å. However, it is noteworthy that current methods in this field including ScanNet (Nat Methods 2022) and PeSTo (Nat Commun 2023) directly compared methods using different label definitions without any alignment in their benchmark studies, likely due to the subtle distinctions among these definitions. For instance, the study of PeSTo directly performed comparisons with ScanNet, MaSIF-site, SPPIDER, and PSIVER. Therefore, we followed these previous works, directly comparing GPSite with other protein-protein binding site predictors.

In the revised “Benchmark datasets” section, we have now provided more details for the binding site definitions in different datasets to avoid any potential ambiguity:

“The benchmark datasets for evaluating binding site predictions of DNA, RNA, peptide, ATP, and HEM are constructed from BioLiP”; “A binding residue is defined if the smallest atomic distance between the target residue and the ligand is < 0.5 Å plus the sum of the Van der Waal’s radius of the two nearest atoms”; “Besides, the benchmark dataset of protein-protein binding sites is directly from ^26^, which contains non-redundant transient heterodimeric protein complexes dated up to May 2021. Surface regions that become solvent inaccessible on complex formation are defined as the ground truth protein-binding sites. The benchmark datasets of metal ion (Zn^2+^, Ca^2+^, Mg^2+^ and Mn^2+^) binding sites are directly from ^18^, which contain non-redundant proteins dated up to December 2021 from BioLiP.”

While GPSite demonstrates the potential to surpass state-of-the-art methods in protein binding site prediction, the evidence supporting these claims seems incomplete. The lack of methodological novelty and the unresolved questions in benchmark consistency and interface definition somewhat undermine the confidence in the results. Therefore, it's not entirely clear if the authors have fully achieved their aims as outlined.The work is useful for the field, especially in disease mechanism elucidation and novel drug design. The availability of genome-scale binding residue annotations GPSite offers is a significant advancement. However, the utility of this tool could be hampered by the aforementioned weaknesses unless they are adequately addressed.

RE: We thank the reviewer for acknowledging the advancement and value of our work, as well as pointing out areas where improvements can be made. As discussed above, we have now carried out the corresponding revisions in the revised manuscript to enhance the completeness and clearness of our work.

**Reviewer #2 (Public Review):**
Summary:This work provides a new framework, "GPsite" to predict DNA, RNA, peptide, protein, ATP, HEM, and metal ions binding sites on proteins. This framework comes with a webserver and a database of annotations. The core of the model is a Geometric featurizer neural network that predicts the binding sites of a protein. One major contribution of the authors is the fact that they feed this neural network with predicted structure from ESMFold for training and prediction (instead of native structure in similar works) and a high-quality protein Language Model representation. The other major contribution is that it provides the public with a new light framework to predict protein-ligand interactions for a broad range of ligands.The authors have demonstrated the interest of their framework with mostly two techniques: ablation and benchmark.Strengths:The performance of this framework as well as the provided dataset and web server make it useful to conduct studies.The ablations of some core elements of the method, such as the protein Language Model part, or the input structure are very insightful and can help convince the reader that every part of the framework is necessary. This could also guide further developments in the field. As such, the presentation of this part of the work can hold a more critical place in this work.

RE: We thank the reviewer for recognizing the contributions of our work and for noting that our experiments are thorough.

Weaknesses:Overall, we can acknowledge the important effort of the authors to compare their work to other similar frameworks. Yet, the lack of homogeneity of training methods and data from one work to the other makes the comparison slightly unconvincing, as the authors pointed out. Overall, the paper puts significant effort into convincing the reader that the method is beating the state of the art. Maybe, there are other aspects that could be more interesting to insist on (usability, interest in protein engineering, and theoretical works).

RE: We sincerely appreciate the reviewer for the constructive and insightful comments. As to the concern of training data heterogeneity raised by the reviewer, it is noteworthy that current studies in this field, such as ScanNet (Nat Methods 2022) and PeSTo (Nat Commun 2023), directly compare methods trained on different datasets in their benchmark experiments. Therefore, we have adhered to the paradigm in these previous works. According to the detailed recommendations by the reviewer, we have now improved our manuscript by incorporating additional ablation studies regarding the effects of training procedure and language model representations, as well as case studies regarding the predicted structure’s quality and GPSite-based function annotations. We have also refined the Discussion section to focus more on the achievements of this work. A comprehensive point-by-point response to the reviewer’s recommendations is provided below.

**Reviewer #2 (Recommendations For The Authors):**
Major comments:Overall I think the work is slightly deserved by its presentation. Some improvements could be made to the paper to better highlight the significance of your contribution.

RE: We thank the reviewer for recognizing the significance of our work!

Line 188: "As expected, the performance of these methods mostly decreases substantially utilizing predicted structures for testing because they were trained with high-quality native structures.This is a major ablation that was not performed in this case. You used the predicted structure to train, while the other did not. One better way to assess the interest of this approach would be to compare the performance of a network trained with only native structure to compare the leap in performance with and without this predicted structure as you did after to assess the interest of some other aspect of your method such as single to multitask.

RE: We thank the reviewer for the valuable recommendation. We have now assessed the benefit of training with predicted instead of native structures, which brings an average AUPR increase of 4.2% as detailed in Appendix 1-note 5 and Appendix 2-table 9. For convenience, we also attach the note and table here:

“We examined the performance under different training and evaluation settings as shown in Appendix 2-table 9. As expected, the model yields exceptional performance (average AUPR of 0.656) when trained and evaluated using native structures. However, if this model is fed with predicted structures of the test proteins, the performance substantially declines to an average AUPR of 0.573. This trend aligns with the observations for other structure-based methods as illustrated in Figure 2. More importantly, in the practical scenario where only predicted structures are available for the target proteins, training the model with predicted structures (i.e., GPSite) results in superior performance than training the model with native structures (average AUPR of 0.594 against 0.573), probably owing to the consistency between the training and testing data. For completeness, the results in Appendix 3-figure 2 are also included where GPSite is tested with native structures (average AUPR of 0.637).”

**Author response table 2. sa4table2:** Performance comparison on the ten binding site test sets under different training and evaluation settings.

Setting	DNA	RNA	Pep	Pro	ATP	HEM	Zn^2+^	Ca^2+^	Mg^2+^	Mn^2+^	Avg
Train: nativeTest: native	0.587	0.634	0.368	0.552	0.746	0.846	0.905	0.705	0.428	0.786	0.656
Train: nativeTest: predicted	0.497	0.554	0.311	0.459	0.704	0.784	0.826	0.546	0.352	0.694	0.573
Train: predictedTest: native	0.554	0.610	0.371	0.529	0.733	0.844	0.890	0.660	0.415	0.761	0.637
Train: predictedTest: predicted(GPSite)	0.516	0.573	0.345	0.484	0.714	0.802	0.859	0.565	0.370	0.709	0.594

Note: The numbers in this table are AUPR values. “Pep” and “Pro” denote peptide and protein, respectively. “Avg” means the average AUPR values among the ten test sets. “native” and “predicted” denote applying native and predicted structures as input, respectively.

Line 263: "ProtTrans consistently obtains competitive or superior performance compared to the MSA profiles, particularly for the target proteins with few homologous sequences (Neff < 2)."This seems a bit far-fetched. If we see clearly in the figure that the performances are far superior for Neff < 2. The performances seem rather similar for higher Neff. Could the author evaluate numerically the significance of the improvement? MSA profiles outperform GPSite on 4 intervals and I don't know the distribution of the data.

RE: We thank the reviewer for the valuable suggestion. We have now revised this sentence to avoid any potential ambiguity:

“As evidenced in Figure 4B and Appendix 2-table 8, ProtTrans consistently obtains competitive or superior performance compared to the MSA profile. Notably, for the target proteins with few homologous sequences (Neff < 2), ProtTrans surpasses MSA profile significantly with an improvement of 3.9% on AUC (P-value = 4.3×10-8).”

The detailed significance tests and data distribution are now added in Appendix 2-table 8 and attached below as Author response table 3 for convenience:

**Author response table 3. sa4table3:** Performance comparison between GPSite and the baseline model using MSA profile for proteins with different Neff values in the combined test set of the ten ligands.

Neff	Sequences	Residues	MSA AUC	GPSite AUC	*P*-value
[1,2)	67	18236	0.818	0.850	4.3×10^-8^
[2,3)	32	9395	0.856	0.854	0.72
[3,4)	71	18328	0.895	0.894	0.13
[4,5)	133	30392	0.901	0.896	4.0×10^-4^
[5,6)	182	39858	0.909	0.916	9.8×10^-4^
[6,7)	226	60128	0.915	0.913	0.10
[7,8)	257	92791	0.920	0.931	1.1×10^-9^
[8,+∞)	947	334455	0.919	0.935	7.0×10^-10^

Note: Significance tests are performed following the procedure in ^12,25^. If *P*-value < 0.05, the difference between the performance is considered statistically significant.

Note: Significance tests are performed following the procedure in Yan and Kurgan, 2017; Xia et al., 2021. If p-value <0.05, the difference between the performance is considered statistically significant.

Line 285: "We first visualized the distributions of residues in this dataset using t-SNE, where the residues are encoded by raw feature vectors encompassing ProtTrans embeddings and DSSP structural properties, or latent embedding vectors from the shared network of GPSite. "Wouldn't embedding from single-task be more relevant to show the interest of multi-task training here? Is the difference that big when comparing embeddings from single-task training to embeddings from multi-task training? Otherwise, I think the evidence from Figure 4e is sufficient, the interest of multitasking could be well-shown by single-task vs. multi-task AUPR and a few examples or predictions that are improved.

RE: We thank the reviewer for the comment. In the second paragraph of the “The effects of protein features and model designs” section, we have compared the performance of multi-task and single-task learning. However, the visualization results in Figure 4D are related to the third paragraph, where we conducted a downstream exploration of the possibility to extend GPSite to other unseen ligands. This is based on the hypothesis that the shared network in GPSite may have captured certain common ligand-binding mechanisms during the preceding multi-task training process. We visualized the distributions of residues in an unseen carbohydrate-binding site dataset using t-SNE, where the residues are encoded by raw feature vectors (ProtTrans and DSSP), or latent embedding vectors from the shared network trained before. Although the shared network has not been specifically trained on the carbohydrate dataset, the latent representations from GPSite effectively improve the discriminability between the binding and non-binding residues as shown in Figure 4D. This finding indicates that the shared network trained on the initial set of ten molecule types has captured common binding mechanisms and may be applied to other unseen ligands.

We have now added more descriptions in this paragraph to avoid potential ambiguity:

“Residues that are conserved during evolution, exposed to solvent, or inside a pocket-shaped domain are inclined to participate in ligand binding. During the preceding multi-task training process, the shared network in GPSite should have learned to capture such common binding mechanisms. Here we show how GPSite can be easily extended to the binding site prediction for other unseen ligands by adopting the pre-trained shared network as a feature extractor. We considered a carbohydrate-binding site dataset from ^54^ which contains 100 proteins for training and 49 for testing. We first visualized the distributions of residues in this dataset using t-SNE ^55^, where the residues are encoded by raw feature vectors encompassing ProtTrans embeddings and DSSP structural properties, or latent embedding vectors from the shared network of GPSite trained on the ten molecule types previously.”

Line291: "Employing these informative hidden embeddings as input features to train a simple MLP exhibits remarkable performance with an AUC of 0.881 (Figure 4E), higher than that of training a single-task version of GPSite from scratch (AUC of 0.853) or other state-of-the-art methods such as MTDsite and SPRINT-CBH."Is it necessary to introduce other methods here? The single-task vs multi-task seems enough for what you want to show?

RE: We thank the reviewer for the comment. As discussed above, here we aim to show the potential of GPSite for the binding site prediction of unseen ligand (i.e., carbohydrate) by adopting the pre-trained shared network as a feature extractor. Thus, we think it’s reasonable to also include the performance of other state-of-the-art methods in this carbohydrate benchmark dataset as baselines.

Line 321: "Specifically, a protein-level binding score can be generated for each ligand by averaging the top *k* predicted scores among all residues. Empirically, we set *k* to 5 for metal ions and 10 for other ligands, considering that the binding interfaces of metal ions are usually smaller."Since binding sites are usually not localized on one single amino-acid, we can expect that most of the top *k* residues are localized around the same area of the protein both spatially and along the sequence. Is it something you observe and could consider in your method?

RE: We thank the reviewer for the comment. We employed a straightforward method (top-*k* average) to convert GPSite’s residue-level annotations into protein-level annotations, where *k* was set empirically based on the distributions of the numbers of binding residues per sequence observed in the training set. We have not put much effort in optimizing this strategy since it mainly serves as a proof-of-concept experiment (Figure 5 A-C) to show the potential of GPSite in discriminating ligand-binding proteins. We have now revised this sentence to better explain how we selected *k*:

“Specifically, a protein-level binding score indicating the overall binding propensity to a specific ligand can be generated by averaging the top *k* predicted scores among all residues. Empirically, we set *k* to 5 for metal ions and 10 for other ligands, considering the distributions of the numbers of binding residues per sequence observed in the training set.”

As for the question raised by the reviewer, we can indeed expect that most of the top *k* predicted binding residues tend to cluster into several but not necessarily one area. For instance, certain macromolecules like DNA may interact with several protein surface patches due to their elongated structures (e.g., Author response image 1). Another case may be a protein binding to multiple molecules of the same ligand type (e.g., Author response image 1).

**Author response image 1. sa4fig1:** The structures of 4XQK (A) and 4KYW (B) in PDB.

Line 327: The accuracy of the GPSite protein-level binding scores is further validated by the ROC curves in Figure 5B, where GPSite achieves satisfactory AUC values for all ligands except protein (AUC of 0.608).Here may be a good place to compare yourself with others, do other frameworks experience the same problem? If so, AUC and AUPR are not relevant here, can you expose some recall scores for example?

RE: We thank the reviewer for the valuable recommendation. We have conducted comprehensive method comparisons in the preceding “GPSite outperforms state-of-the-art methods” section, where GPSite surpasses all existing frameworks across various ligands. Here, the genome-wide analyses of Swiss-Prot in Figure 5 serve as a downstream demonstration of GPSite’s capacity for large-scale annotations. We didn’t compare with other methods since most of them are time-consuming or memory-consuming, thus unavailable to process sequences of substantial quantity or length. For example, it takes about 8 min for the MSA-based method GraphBind to annotate a protein with 500 residues, while it just takes about 20 s for GPSite (see Appendix 3-figure 1 for detailed runtime comparison). It is also challenging for the atom-graph-based method PeSTo to process structures more than 100 kDa (~1000 residues) on a 32 GB GPU as the authors suggested, while GPSite can easily process structures containing up to 2500 residues on a 16 GB GPU.

Regarding the recall score mentioned by the reviewer, GPSite achieves a recall of 0.95 (threshold = 0.5) for identifying protein-binding proteins. This indicates that GPSite can accurately identify positive samples, but it also tends to misclassify negative samples as positive. In our original manuscript, we claimed that “This may be ascribed to the fact that protein-protein interactions are ubiquitous in living organisms while the Swiss-Prot function annotations are incomplete”. To better support this claim, we have now added two examples in Appendix 1-note 7, where GPSite confidently predicted the presences of the “protein binding” function (GO:0005515). Notably, this function was absent in these two proteins in the Swiss-Prot database at the time of manuscript preparation (release: 2023-05-03), but has been included in the latest release of Swiss-Prot (release: 2023-11-08). For convenience, we also attach the note here:

“As depicted in Figure 5A, GPSite assigns relatively high prediction scores to the proteins without “protein binding” function in the Swiss-Prot annotations, leading to a modest AUC value of 0.608 (Figure 5B). This may be ascribed to the fact that protein-protein interactions are ubiquitous in living organisms while the Swiss-Prot function annotations are incomplete. To support this hypothesis, we present two proteins as case studies, both sharing < 20% sequence identity with the protein-binding training set of GPSite. The first case is Aminodeoxychorismate synthase component 2 from *Escherichia coli* (UniProt ID: P00903). GPSite confidently predicted this protein as a protein-binding protein with a high prediction score of 0.936. Notably, this protein was not annotated with the “protein binding” function (GO:0005515) or any of its GO child terms in the Swiss-Prot database at the time of manuscript preparation (https://rest.uniprot.org/unisave/P00903?format=txt&versions=171, release: 2023-05-03). However, in the latest release of Swiss-Prot (https://rest.uniprot.org/unisave/P00903?format=txt&versions=174, release: 2023-11-08) during manuscript revision, this protein is annotated with the “protein heterodimerization activity” function (GO:0046982), which is a child term of “protein binding”. In fact, the heterodimerization activity of this protein has been validated through experiments in the year of 1996 (PMID: 8679677), indicating the potential incompleteness of the Swiss-Prot annotations. The other case is Hydrogenase-2 operon protein HybE from *Escherichia coli* (UniProt ID: P0AAN1), which was also predicted as a protein-binding protein by GPSite (score = 0.909). Similarly, this protein was not annotated with the “protein binding” function in the Swiss-Prot database at the time of manuscript preparation (https://rest.uniprot.org/unisave/P0AAN1?format=txt&versions=108). However, in the latest release of Swiss-Prot (https://rest.uniprot.org/unisave/P0AAN1?format=txt&versions=111), this protein is annotated with the “preprotein binding” function (GO:0070678), which is a child term of “protein binding”. In fact, the preprotein binding function of this protein has been validated through experiments in the year of 2003 (PMID: 12914940). These cases demonstrate the effectiveness of GPSite for completing the missing function annotations in Swiss-Prot.”

Line 381: 'Despite the noteworthy advancements achieved by GPSite, there remains scope for further improvements. Given that the ESM Metagenomic Atlas ^34^ provides 772 million predicted protein structures along with pre-computed language model embeddings, self-supervised learning can be employed to train a GPSite model for predicting masked sequence and structure attributes, or maximizing the similarity between the learned representations of substructures from identical proteins while minimizing the similarity between those from different proteins using a contrastive loss function training from scratch. Additional opportunities for upgrade exist within the network architecture. For example, a variational Expectation-Maximization (EM) framework ^58^ can be adopted to handle the hierarchical graph structure inherent in proteins, which contains the top view of the residue graph and the bottom view of the atom graph inside a residue. Such an EM procedure enables training two separate graph neural networks for the two views while simultaneously allowing interaction and mutual enhancement between the two modules. Meta-learning could also be explored in this multi-task scenario, which allows fast adaptation to unseen tasks with limited labels.'I think this does not belong here. It feels like half of your discussion is not talking about the achievements of this paper but future very specific directions. Focus on the take-home arguments (performances of the model, ability to predict a large range of tasks, interest in key components of your model, easy use) of the paper and possible future direction but without being so specific.

RE: We thank the reviewer for the valuable suggestion. We have now simplified the discussions on the future directions notably:

“Despite the noteworthy advancements achieved by GPSite, there remains scope for further improvements. GPSite may be improved by pre-training on the abundant predicted structures in ESM Metagenomic Atlas, and then fine-tuning on binding site datasets. Besides, the hidden embeddings from ESMFold may also serve as informative protein representations. Additional opportunities for upgrade exist within the network architecture. For example, a variational Expectation-Maximization framework can be adopted to handle the hierarchical atom-to-residue graph structure inherent in proteins. Meta-learning could also be explored in this multi-task scenario, which allows fast adaptation to unseen tasks with limited labels.”

Overall there is also a lack of displayed structure. You should try to select a few examples of binding sites that were identified correctly by your method and not by others, if possible get some insights on why. Also, some negative examples could be interesting so as to have a better idea of the interest.

RE: We thank the reviewer for the valuable recommendation. We have performed a case study for the structure of the glucocorticoid receptor in Figure 3 D-H to illustrate a potential reason for the robustness of GPSite. Moreover, we have now added a case study in Appendix 1-note 3 and Appendix 3-figure 5 to explain why GPSite sometimes is not as accurate as the state-of-the-art structure-based method. For convenience, we also attach the note and figure here:

“Here we present an example of an RNA-binding protein, i.e., the ribosome biogenesis protein ERB1 (PDB: 7R6Q, chain m), to illustrate the impact of predicted structure’s quality. As shown in Appendix 3-figure 5, ERB1 is an integral component of a large multimer structure comprising protein and RNA chains (i.e., the state E2 nucleolar 60S ribosome biogenesis intermediate). Likely due to the neglect of interactions from other protein chains, ESMFold fails to predict the correct conformation of the ERB1 chain (TM-score = 0.24). Using this incorrect predicted structure, GPSite achieves an AUPR of 0.580, lower than GraphBind input with the native structure (AUPR = 0.636). However, the performance of GraphBind substantially declines to an AUPR of 0.468 when employing the predicted structure as input. Moreover, if GPSite adopts the native structure for prediction, a notable performance boost can be obtained (AUPR = 0.681).”

**Author response image 2. sa4fig2:** The prediction results of GPSite and GraphBind for the ribosome biogenesis protein ERB1. (A) The state E2 nucleolar 60S ribosome biogenesis intermediate (PDB: 7R6Q). The ribosome biogenesis protein ERB1 (chain m) is highlighted in blue, while other protein chains are colored in gray. The RNA chains are shown in orange. (B) The RNA-binding sites on ERB1 (colored in red). (C) The ESMFold-predicted structure of ERB1 (TM-score = 0.24). The RNA-binding sites are also mapped onto this predicted structure (colored in red). (D-G) The prediction results of GPSite and GraphBind for the predicted and native ERB1 structures. The confidence of the predictions is represented with a gradient of color from blue for non-binding to red for binding.

Minor comments:Line 169: "Note that since our test sets may partly overlap with the training sets of these methods, the results reported here should be the upper limits for the existing methods."Yes, but they were potentially not trained on the most recent structures in that case. These methods could also see improved performance with an updated training set.

RE: We thank the reviewer for the comment. We have now deleted this sentence.

Line176: "Since 358 of the 375 proteins in our protein-binding site test set share > 30% identity with the training sequences of PeSTo, we re-split our protein-binding dataset to generate a test set of 65 proteins sharing < 30% identity with the training set of PeSTo for a fair evaluation."Too specific to be here in my opinion.

RE: We thank the reviewer for the comment. We have now moved these details to Appendix 1-note 2. The description in the main text here is now more concise:

“Given the substantial overlap between our protein-binding site test set and the training set of PeSTo, we conducted separate training and comparison using the datasets of PeSTo, where GPSite still demonstrates a remarkable improvement over PeSTo (Appendix 1-note 2).”

Figure 2. The authors should try to either increase Fig A's size or increase the font size. This could probably be done by compressing the size of Figure C into a single figure.

RE: We thank the reviewer for the suggestion. We have now increased the font size in Figure A. Besides, the figures in the final version of the manuscript should be clearer where we could upload SVG files.

Have you tried using embeddings from more structure-aware pLM such as ESMFold embeddings (fine-tuned) or Pro**s**tTrans (that may be more recent than this study)?

RE: We thank the reviewer for the insightful comment. We have not yet explored the embeddings from structure-aware pLM, but we acknowledge its potential as a promising avenue for future investigation. We have now added this point in our Discussion section:

“Besides, the hidden embeddings from ESMFold may also serve as informative protein representations.”

**Reviewer #3 (Public Review):**
SummaryThe authors of this work aim to address the challenge of accurately and efficiently identifying protein binding sites from sequences. They recognize that the limitations of current methods, including reliance on multiple sequence alignments or experimental protein structure, and the under-explored geometry of the structure, which limit the performance and genome-scale applications. The authors have developed a multi-task network called GPSite that predicts binding residues for a range of biologically relevant molecules, including DNA, RNA, peptides, proteins, ATP, HEM, and metal ions, using a combination of sequence embeddings from protein language models and ESMFold-predicted structures. Their approach attempts to extract residual and relational geometric contexts in an end-to-end manner, surpassing current sequence-based and structure-based methods.StrengthsThe GPSite model's ability to predict binding sites for a wide variety of molecules, including DNA, RNA, peptides, and various metal ions.Based on the presented results, GPSite outperforms state-of-the-art methods in several benchmark datasets.GPSite adopts predicted structures instead of native structures as input, enabling the model to be applied to a wider range of scenarios where native structures are rare.The authors emphasize the low computational cost of GPSite, which enables rapid genome-scale binding residue annotations, indicating the model's potential for large-scale applications.

RE: We thank the reviewer for recognizing the significance and value of our work!

WeaknessesOne major advantage of GPSite, as claimed by the authors, is its efficiency. Although the manuscript mentioned that the inference takes about 5 hours for all datasets, it remains unclear how much improvement GPSite can offer compared with existing methods. A more detailed benchmark comparison of running time against other methods is recommended (including the running time of different components, since some methods like GPSite use predicted structures while some use native structures).

RE: We thank the reviewer for the valuable suggestion. Empirically, it takes about 5-20 min for existing MSA-based methods to make predictions for a protein with 500 residues, while it only takes about 1 min for GPSite (including structure prediction). However, it is worth noting that some predictors in our benchmark study are solely available as webservers, and it is challenging to compare the runtime between a standalone program and a webserver due to the disparity in hardware configurations. Therefore, we have now included comprehensive runtime comparisons between the GPSite webserver and other top-performing servers in Appendix 3-figure 1 to illustrate the practicality and efficiency of our method. For convenience, we also attach the figure here as Author response image 3. The corresponding description is now added in the “GPSite outperforms state-of-the-art methods” section:

“Moreover, GPSite is computationally efficient, achieving comparable or faster prediction speed compared to other top-performing methods (Appendix 3-figure 1).”

**Author response image 3. sa4fig3:** Runtime comparison of the GPSite webserver with other top-performing servers. Five protein chains (i.e., 8HN4_B, 8USJ_A, 8C1U_A, 8K3V_A and 8EXO_A) comprising 100, 300, 500, 700, and 900 residues, respectively, were selected for testing, and the average runtime is reported for each method. Note that a significant portion of GPSite’s runtime (75 s, indicated in orange) is allocated to structure prediction using ESMFold.

Since the model uses predicted protein structure, the authors have conducted some studies on the effect of the predicted structure's quality. However, only the 0.7 threshold was used. A more comprehensive analysis with several different thresholds is recommended.

RE: We thank the reviewer for the comment. We assessed the effect of the predicted structure's quality by evaluating GPSite’s performance on high-quality (TM-score > 0.7) and low-quality (TM-score ≤ 0.7) predicted structures. We did not employ multiple thresholds (e.g., 0.3, 0.5, and 0.7), as the majority of proteins in the test sets were accurately predicted by ESMFold. Specifically, as shown in Figure 3B, Appendix 3-figure 3 and Appendix 2-table 5, the numbers of proteins with TM-score ≤ 0.7 are small in most datasets (e.g., 42 for DNA and 17 for ATP). Consequently, there is insufficient data available for analysis with lower thresholds, except for the RNA test set. Notably, Figure 3C presents a detailed inspection of the 104 proteins with TM-score < 0.5 in the RNA test set. Within this subset, GPSite consistently outperforms the state-of-the-art structure-based method GraphBind with predicted structures as input, regardless of the prediction quality of ESMFold. Only in cases where structures are predicted with extremely low quality (TM-score < 0.3) does GPSite fall behind GraphBind input with native structures. This result further demonstrates the robustness of GPSite. We have now added clearer explanations in the “GPSite is robust for low-quality predicted structures” section:

“Figure 3B and Appendix 3-figure 3 show the distributions of TM-scores between native and predicted structures calculated by US-align in the ten benchmark datasets, where most proteins are accurately predicted with TM-score > 0.7 (see also Appendix 2-table 5)”; “Given the infrequency of low-quality predicted structures except for the RNA test set, we took a closer inspection of the 104 proteins with predicted structures of TM-score < 0.5 in the RNA test set.”

To demonstrate the robustness of GPSite, the authors performed a case study on human GR containing two zinc fingers, where the predicted structure is not perfect. The analysis could benefit from more a detailed explanation of why the model can still infer the binding site correctly even though the input structural information is slightly off.

RE: We thank the reviewer for the comment. We have actually explained the potential reason for the robustness of GPSite in the second paragraph of the “GPSite is robust for low-quality predicted structures” section. In summary, although the whole structure of this protein is not perfectly predicted, the local structures of the binding domains of peptide, DNA and Zn^2+^ are actually predicted accurately as evidenced by the superpositions of the native and predicted structures in Figure 3D and 3E. Therefore, GPSite can still make reliable predictions. We have now revised this paragraph to explain these more clearly:

“Figure 3D shows the structure of the human glucocorticoid receptor (GR), a transcription factor that binds DNA and assembles a coactivator peptide to regulate gene transcription (PDB: 7PRW, chain A). The DNA-binding domain of GR also consists of two C4-type zinc fingers to bind Zn^2+^ ions. Although the structure of this protein is not perfectly predicted (TM-score = 0.72), the local structures of the binding domains of peptide and DNA are actually predicted accurately as viewed by the superpositions of the native and predicted structures in Figure 3D and 3E. Therefore, GPSite can correctly predict all Zn^2+^ binding sites and precisely identify the binding sites of DNA and peptide with AUPR values of 0.949 and 0.924, respectively (Figure 3F, G and H).”

To analyze the relatively low AUC value for protein-protein interactions, the authors claimed that it is "due to the fact that protein-protein interactions are ubiquitous in living organisms while the Swiss-Prot function annotations are incomplete", which is unjustified. It is highly recommended to support this claim by showing at least one example where GPSite's prediction is a valid binding site that is not present in the current Swiss-Prot database or via other approaches.

RE: We thank the reviewer for the valuable recommendation. To support this claim, we have now added two examples in Appendix 1-note 7, where GPSite confidently predicted the presences of the “protein binding” function (GO:0005515). Notably, this function was absent in these two proteins in the Swiss-Prot database at the time of manuscript preparation (release: 2023-05-03), but has been included in the latest release of Swiss-Prot (release: 2023-11-08). For convenience, we also attach the note below:

“As depicted in Figure 5A, GPSite assigns relatively high prediction scores to the proteins without “protein binding” function in the Swiss-Prot annotations, leading to a modest AUC value of 0.608 (Figure 5B). This may be ascribed to the fact that protein-protein interactions are ubiquitous in living organisms while the Swiss-Prot function annotations are incomplete. To support this hypothesis, we present two proteins as case studies, both sharing < 20% sequence identity with the protein-binding training set of GPSite. The first case is Aminodeoxychorismate synthase component 2 from *Escherichia coli* (UniProt ID: P00903). GPSite confidently predicted this protein as a protein-binding protein with a high prediction score of 0.936. Notably, this protein was not annotated with the “protein binding” function (GO:0005515) or any of its GO child terms in the Swiss-Prot database at the time of manuscript preparation (https://rest.uniprot.org/unisave/P00903?format=txt&versions=171, release: 2023-05-03). However, in the latest release of Swiss-Prot (https://rest.uniprot.org/unisave/P00903?format=txt&versions=174, release: 2023-11-08) during manuscript revision, this protein is annotated with the “protein heterodimerization activity” function (GO:0046982), which is a child term of “protein binding”. In fact, the heterodimerization activity of this protein has been validated through experiments in the year of 1996 (PMID: 8679677), indicating the potential incompleteness of the Swiss-Prot annotations. The other case is Hydrogenase-2 operon protein HybE from *Escherichia coli* (UniProt ID: P0AAN1), which was also predicted as a protein-binding protein by GPSite (score = 0.909). Similarly, this protein was not annotated with the “protein binding” function in the Swiss-Prot database at the time of manuscript preparation (https://rest.uniprot.org/unisave/P0AAN1?format=txt&versions=108). However, in the latest release of Swiss-Prot (https://rest.uniprot.org/unisave/P0AAN1?format=txt&versions=111), this protein is annotated with the “preprotein binding” function (GO:0070678), which is a child term of “protein binding”. In fact, the preprotein binding function of this protein has been validated through experiments in the year of 2003 (PMID: 12914940). These cases demonstrate the effectiveness of GPSite for completing the missing function annotations in Swiss-Prot.”

The authors reported that many GPSite-predicted binding sites are associated with known biological functions. Notably, for RNA-binding sites, there is a significantly higher proportion of translation-related binding sites. The analysis could benefit from a further investigation into this observation, such as the analyzing the percentage of such interactions in the training site. In addition, if there is sufficient data, it would also be interesting to see the cross-interaction-type performance of the proposed model, e.g., train the model on a dataset excluding specific binding sites and test its performance on that class of interactions.

RE: We thank the reviewer for the suggestion. We would like to clarify that the analysis in Figure 5C was conducted at “protein-level” instead of “residue-level”. As described in the second paragraph of the “Large-scale binding site annotation for Swiss-Prot” section, a protein-level ligand-binding score was assigned to a protein by averaging the top *k* residue-level predicted binding scores. This protein-level score indicates the overall binding propensity of the protein to a specific ligand. We gathered the top 20,000 proteins with the highest protein-level binding scores for each ligand and found that their biological process annotations from Swiss-Prot were consistent with existing knowledge. We have now revised the corresponding sentence to explain these more clearly:

“Exploiting the residue-level binding site annotations, we could readily extend GPSite to discriminate between binding and non-binding proteins of various ligands. Specifically, a protein-level binding score indicating the overall binding propensity to a specific ligand can be generated by averaging the top *k* predicted scores among all residues.”

As for the cross-interaction-type performance raised by the reviewer, we have now conducted cross-type evaluations to investigate the specificity of the ligand-specific MLPs and the inherent similarities among different ligands in Appendix 1-note 6 and Appendix 2-table 10. For convenience, we also attach the note and table here:

“We conducted cross-type evaluations by applying different ligand-specific MLPs in GPSite for the test sets of different ligands. As shown in Appendix 2-table 10, for each ligand-binding site test set, the corresponding ligand-specific network consistently achieves the best performance. This indicates that the ligand-specific MLPs have specifically learned the binding patterns of particular molecules. We also noticed that the cross-type performance is reasonable for the ligands sharing similar properties. For instance, the DNA-specific MLP exhibits a reasonable AUPR when predicting RNA-binding sites, and vice versa. Similar trends are also observed between peptide and protein, as well as among metal ions as expected. Interestingly, the cross-type performance between ATP and HEM is also acceptable, potentially attributed to their comparable molecular weights (507.2 and 616.5, respectively).”

**Author response table 4. sa4table4:** Cross-type performance by applying different ligand-specific MLPs in GPSite for the test sets of different ligands.

Ligand-specific MLP	Ligand-binding site test set
DNA	RNA	Pep	Pro	ATP	HEM	Zn^2+^	Ca^2+^	Mg^2+^	Mn^2+^
DNA	**0.516**	0.461	0.158	0.327	0.123	0.425	0.032	0.033	0.028	0.072
RNA	0.381	**0.573**	0.170	0.332	0.189	0.549	0.038	0.049	0.037	0.093
Pep	0.170	0.199	**0.345**	0.410	0.089	0.479	0.046	0.027	0.028	0.080
Pro	0.187	0.214	0.201	**0.484**	0.031	0.117	0.030	0.026	0.015	0.025
ATP	0.193	0.319	0.165	0.296	**0.714**	0.762	0.036	0.076	0.062	0.138
HEM	0.231	0.316	0.236	0.321	0.544	**0.802**	0.073	0.026	0.040	0.086
Zn^2+^	0.076	0.164	0.069	0.197	0.077	0.115	**0.859**	0.136	0.111	0.622
Ca^2+^	0.091	0.197	0.079	0.234	0.151	0.074	0.114	**0.565**	0.317	0.460
Mg^2+^	0.117	0.206	0.091	0.232	0.265	0.208	0.192	0.468	**0.370**	0.597
Mn^2+^	0.108	0.196	0.095	0.226	0.245	0.237	0.627	0.390	0.321	**0.709**

Note: “Pep” and “Pro” denote peptide and protein, respectively. The numbers in this table are AUPR values. The best/second-best result in each test set is indicated by bold/underlined font.